# Cellular and circuit features distinguish mouse dentate gyrus semilunar granule cells and granule cells activated during contextual memory formation

Laura Dovek[1,2], Mahboubeh Ahmadi[2], Krista Marrero[2], Edward Zagha[1,3,4], Vijayalakshmi Santhakumar[1,2,3]*

[1]Biomedical Sciences Graduate Program, University of California Riverside, Riverside, United States; [2]Department of Molecular, Cell and Systems Biology, University of California Riverside, Riverside, United States; [3]Neuroscience Graduate Program, University of California Riverside, Riverside, United States; [4]Department of Psychology, University of California Riverside, Riverside, United States

## eLife Assessment

This **useful** study describes distinctive characteristics of dentate gyrus granule cells and semilunar cells that are recruited during contextual memory processing. The study provides **solid** evidence to suggest mechanisms that may be involved in the recruitment of neurons into memory engrams in the dentate gyrus.

*For correspondence: vijayas@ucr.edu

**Abstract** The dentate gyrus is critical for spatial memory formation and shows task-related activation of cellular ensembles considered as memory engrams. Semilunar granule cells (SGCs), a sparse dentate projection neuron subtype, were reported to be enriched among behaviorally activated neurons. By examining SGCs and granule cells (GCs) labeled during contextual memory formation in TRAP2 mice, we empirically tested competing hypotheses for GC and SGC recruitment into memory ensembles. Consistent with more excitable neurons being recruited into memory ensembles, SGCs showed greater sustained firing than GCs. Additionally, labeled SGCs showed less adapting firing than unlabeled SGCs. The lack of glutamatergic connections between behaviorally labeled SGCs and GCs in our recordings is inconsistent with SGC-driven local circuit feedforward excitation underlying ensemble recruitment. Moreover, there was little evidence for individual SGCs or labeled neuronal ensembles supporting lateral inhibition of unlabeled neurons. Instead, labeled GCs and SGCs received more spontaneous excitatory synaptic inputs than their unlabeled counterparts. Labeled neuronal pairs received more temporally correlated spontaneous excitatory synaptic inputs than labeled-unlabeled neuronal pairs. These findings challenge the proposal that SGCs drive dentate GC ensemble refinement, while supporting a role for intrinsic excitability and correlated inputs in preferential SGC recruitment to contextual memory engrams.

## Introduction

The ability of neural circuits to represent unique experiences and events as distinct neuronal representations that can be recalled and updated is fundamental to memory formation. The hippocampal dentate gyrus (DG) is considered central for both novelty detection and the formation of episodic

memories (*Hunsaker et al., 2008*; *Liu et al., 2012*; *Hainmueller and Bartos, 2020*; *Danieli et al., 2023*). The DG receives dense information from diverse cortical regions through the perforant path projections from the entorhinal cortex (*van Groen et al., 2003*). Yet, relatively few of the numerous closely packed dentate projection neurons, granule cells (GCs), are activated and engage downstream hippocampal circuits. This sparsening of activity is proposed as critical for pattern separation, a process by which the DG helps disambiguate similar memories (*McHugh et al., 2007*; *Hainmueller and Bartos, 2018*). Still, the mechanisms that govern how select subsets of neurons are activated during memory formation are not fully understood.

The cellular representations of memories, known as *engrams,* refer to distinct groups of neurons activated during memory acquisition (*Semon, 1909*; *Josselyn et al., 2015*; *Josselyn and Tonegawa, 2020*). Recently, a sparse subset of DG projection neurons, known as semilunar granule cells (SGCs), has been found to be overrepresented among neurons labeled by the expression of the activity-dependent immediate early gene (IEG) c-*Fos* during hippocampus-dependent behaviors in TRAP2 reporter mice (*Erwin et al., 2020*). SGCs, like GCs, have molecular layer dendrites and project axons to CA3 (*Williams et al., 2007*; *Afrasiabi et al., 2022*). However, unbiased cluster analyses of morphometric data have revealed that structural features can reliably distinguish SGCs from GCs based on their wider dendritic arbor, greater soma width to length ratio, and more numerous primary dendrites (*Williams et al., 2007*; *Gupta et al., 2020*; *Afrasiabi et al., 2022*). Despite SGCs being estimated to make up only ~3% of the total GC population (*Save et al., 2019*), their preferential activation in memory tasks suggests that SGCs may possess unique physiology or connectivity to support recruitment to engrams. However, why SGCs may be preferentially recruited, and whether they shape DG ensemble refinement, remains unresolved.

There are complementary theories for why certain neurons are selectively activated during memory formation and for how the active cell ensembles may be refined by circuit processes. One hypothesis is that neurons are recruited to memory ensembles based on greater excitability (*Yiu et al., 2014*; *Gouty-Colomer et al., 2016*). According to this hypothesis, distinct cohorts of neurons may have higher intrinsic excitability during certain periods; this propensity biases them to fire preferentially in response to inputs and to be recruited into behaviorally activated ensembles (*Yiu et al., 2014*). Relatedly, it has been suggested that newborn GCs are preferentially recruited to engrams because of their higher excitability (*Kee et al., 2007*). However, it is not known whether intrinsic physiological features of SGCs, which show sustained afferent-driven firing (*Larimer and Strowbridge, 2010*; *Afrasiabi et al., 2022*), support their disproportionate representation among behaviorally activated DG ensembles.

In addition to intrinsic properties, neuronal recruitment can be refined by local circuit feedforward or recurrent excitation. One possibility is that glutamatergic interconnectivity aids in engram refinement. Indeed, reports of higher connection probability and strengthening of excitatory synapses between GCs and CA3 pyramidal cells labeled based on IEG expression following fear conditioning (*Ryan et al., 2015*) support this possibility. Although GCs typically do not innervate other GCs, SGCs have axon collaterals in the molecular layer (*Williams et al., 2007*; *Save et al., 2019*), which positions them to potentially form synaptic contacts with GCs. However, whether SGCs directly activate GCs and whether SGCs refine their recruitment to behaviorally active neuronal ensembles remain to be tested. Evaluation of synaptic connectivity between neuronal pairs in ensembles labeled based on IEG expression during memory formation would allow us to test whether recurrent glutamatergic connections support DG ensemble recruitment. Simultaneously, since connectivity between SGCs and GCs is likely to be sparse, this experimental paradigm allows us to address the open question of whether SGCs synaptically activate GCs.

A leading hypothesis for DG circuit refinement of behaviorally active neuronal ensembles, particularly in the context of pattern separation, is through lateral feedback inhibition of surrounding GCs (*Walker et al., 2010*; *Cayco-Gajic and Silver, 2019*; *Guzman et al., 2021*; *Borzello et al., 2023*). The characteristic robust feedback inhibition in the DG holds promise as a mechanism by which activated engram neurons recruit interneurons to selectively inhibit surrounding neurons (*Espinoza et al., 2018*). However, this is difficult to reconcile with the exceedingly sparse GC-mediated lateral inhibition in recordings from GC pairs (*Espinoza et al., 2018*; *Braganza et al., 2020*). It is possible that neurons recruited during contextual memory formation undergo synaptic refinement for better recruitment of lateral inhibition when compared to a naïve circuit. Alternatively, lateral inhibition by

neurons in a memory-related ensemble may be largely driven by the recruited SGC populations. SGCs are ideally poised to mediate this effect as their axon collaterals have been shown to form perisomatic synapses on parvalbumin-expressing fast-spiking basket cells known for their feedback inhibition of GCs (*Rovira-Esteban et al., 2020*). Consistent with a role for SGCs in supporting feedback inhibition, afferent-evoked persistent firing in SGCs is correlated with sustained basket cell and hilar interneuron firing and prolonged inhibitory synaptic barrages in GCs and SGCs (*Larimer and Strowbridge, 2010*; *Afrasiabi et al., 2022*). While focal optogenetic activation of a random population of virally labeled GCs elicits robust inhibition in surrounding GCs (*Stefanelli et al., 2016*), whether the sparse neuronal populations activated during behaviorally driven encoding mediate lateral inhibition of surrounding neurons remains to be tested.

Finally, it is reasonable to posit that precise connectivity of afferent inputs determines downstream activation of a sparse population of DG neurons. Indeed, there is evidence for input-dependent recruitment of neuronal cohorts in the amygdala during fear conditioning (*Gouty-Colomer et al., 2016*). However, whether shared inputs constrain coactivation of neurons, and whether input specificity acts in concert with intrinsic and circuit features to determine which GCs and SGCs are activated, remains unknown. While several studies have focused on DG engram formation (*Liu et al., 2012*; *Ryan et al., 2015*), only recently has there been an attempt to explicitly distinguish GCs from SGCs (*Erwin et al., 2020*). Thus, the specific circuit mechanisms underlying behaviorally relevant DG ensemble refinement during memory encoding and roles of SGCs remain to be determined.

Here, we used TRAP2 transgenic mice for c-*Fos*-driven labeling of DG ensembles during behavioral tasks (*Guenthner et al., 2013*; *DeNardo et al., 2019*) to label active DG ensembles and undertook ex vivo dual patch clamp and optogenetic recordings in morphologically characterized GCs and SGCs. We use these data to evaluate competing proposals for refinement of cellular ensemble representations in the DG. We specifically focused on differential recruitment of GCs and SGCs in DG ensembles, the potential roles for SGCs in shaping DG circuit processing and refining neuronal ensembles, and the role of afferent inputs in shaping DG neuronal ensemble recruitment.

## Results

### SGCs are reliably recruited during contextual memory formation

The DG is a primary relay for memory processing (*Amaral et al., 2007*). However, the mechanisms by which memory-related cellular ensembles are selectively activated during memory encoding are not fully understood. To determine the DG-dependent naturalistic behavioral tasks which can recruit a DG ensemble for physiological analysis, we compared the Barnes maze (BM) and an enriched environment (EE) exposure. We were particularly interested in identifying a behavioral context independent of fear conditioning that activated large cohorts of DG neurons, thereby enabling microcircuit analyses via physiological recordings. Behaviorally activated '*engram*' neurons, referred to henceforth as 'labeled neurons', are neurons in TRAP2 mice induced to express the reporter (tdT or ChR2-YFP) downstream of the activity-dependent IEG c-*Fos* during BM or EE. 'Unlabeled neurons' lack reporter expression. Littermate pairs of TRAP2-tdT mice (four pairs) were either trained in the BM spatial learning task or exposed to an EE, tasks known to engage the DG. Mice trained in the BM showed progressive decrease in primary latency and primary errors to locate the escape box (*Figure 1—figure supplement 1*), demonstrating improved performance from acquisition days 1 through 6. Barnes maze unbiased strategy (BUNS) classification and cognitive scores to assess the use of spatial search strategy (*Illouz et al., 2016*) revealed that the mice transitioned from using a random or serial search strategy to a spatial strategy as they progressed through acquisition days (*Figure 1—figure supplement 1*). Both cohorts were induced with tamoxifen during respective behavioral paradigms, on day 6 of BM acquisition or halfway through the 1-day EE exposure, to label active neurons (*Figure 1A and B*). Comparison of the number of DG c-*Fos*-expressing (tdT-positive) neurons in hippocampal sections from mice 1 week after tamoxifen induction revealed significantly more tdT-labeled neurons following EE exposure than after BM acquisition (*Figure 1C–E*; # of tdT-labeled cells per slice: EE: 33.90±2.13, BM 13.43±0.90, n=40 slices from 4 animals per group, p=0.0409 by nested t-test). Consistent with previous reports in several other hippocampus-dependent tasks (*Erwin et al., 2020*), the suprapyramidal (upper) blade of the DG showed more neurons labeled than the infrapyramidal (lower) blade following both BM training and EE exposure (*Figure 1F*). To determine whether tagged neurons show

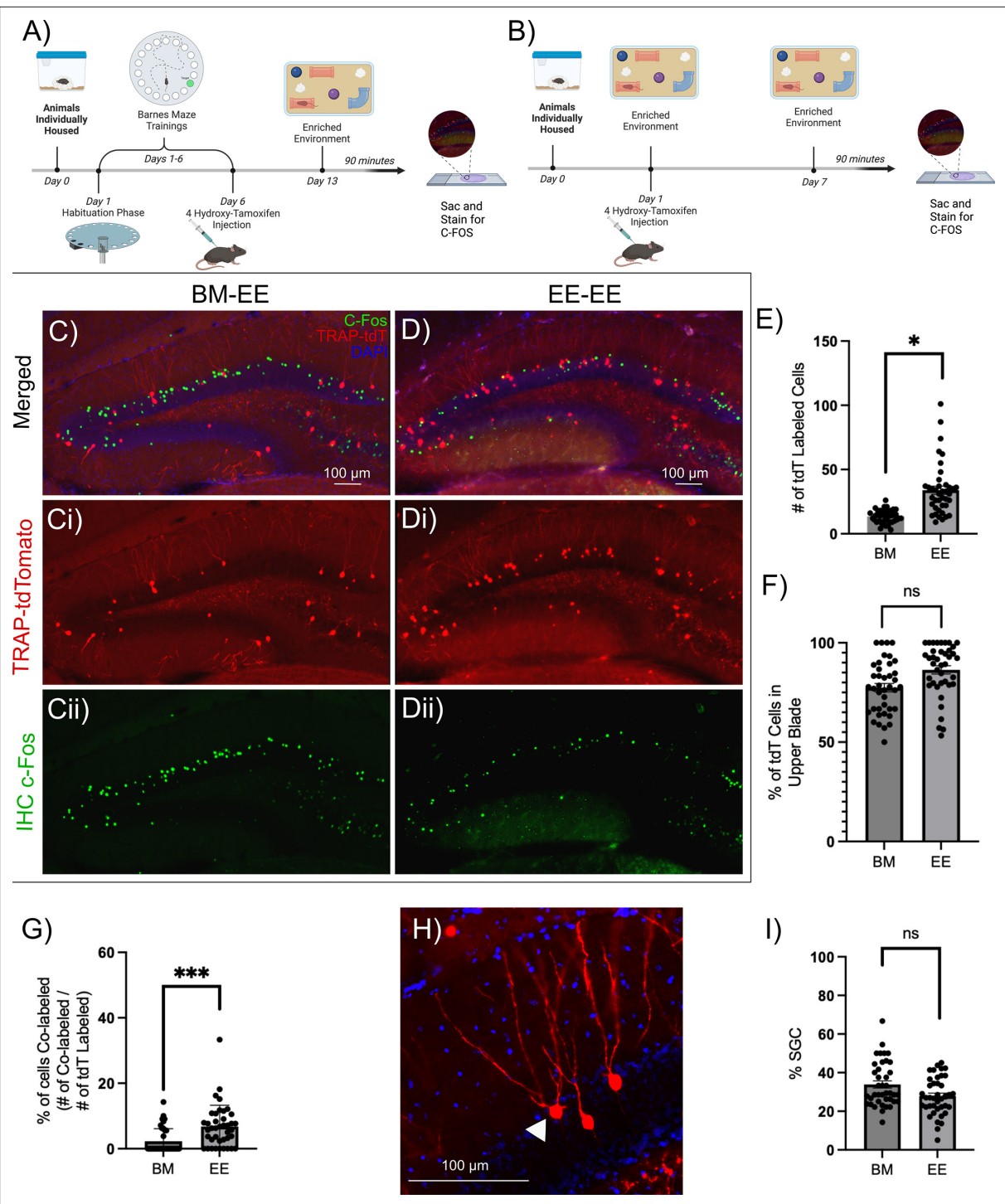

**Figure 1.** Task-associated dentate gyrus (DG) labeled neurons show consistent activation of semilunar granule cells (SGCs) and paradigm-specific reactivation. (**A–B**) Schematic of experimental timeline for animals trained in the Barnes maze (BM) task followed by exposure to enriched environment (EE), the BM-EE cohort (BM) group (**A**) and mice housed in EE followed by reintroduction of EE, the EE-EE cohort (EE) group (**B**), created with BioRender. com. (**C–D**) Representative epifluorescence image of a section from mice 1 week after induction of tdT labeling (Ci, Di) following BM testing (**C**) or EE testing (**D**) and c-*Fos* immunostaining (**Cii, Dii**) following subsequent EE exposure. (**E–F**) Quantification of number of tdT-labeled cells per slice (**E**) and summary of proportion of tdT-labeled cells in the upper blade of the DG per slice (**F**). (**G**) Summary of proportion of tdT cells co-labeled with c-*Fos* (green). (**H**) Representative TRAP-tdT section showing distinct SGC morphology (white arrowhead). (**I**) Plot of % of tdT cells that had morphology consistent with SGCs.Data are presented as mean ± SEM. * indicates p<0.05, *** indicates p=0.0003 by nested t-test, n=4 subjects/treatment.

The online version of this article includes the following source data and figure supplement(s) for figure 1:

*Figure 1 continued on next page*

*Figure 1 continued*

**Source data 1.** Data underlying *Figure 1E, F, G and I*.

**Figure supplement 1.** Search strategies adopted in the Barnes maze task.

**Figure supplement 1—source data 1.** Data underlying *Figure 1—figure supplement 1A-D*.

task-specific reactivation 1 week after induction, mice were exposed to EE prior to perfusion, and sections were immunostained for c-*Fos*. The distribution of neurons immunolabeled for c-*Fos* following EE exposure showed no apparent difference between mice previously exposed to BM followed by EE and those exposed to EE twice (*Figure 1Cii and Dii*). However, consistent with memory-related neuronal tagging, mice with prior exposure to EE showed greater co-labeling of tdT-positive neurons with c-*Fos* immunostaining than mice that were initially trained in the BM task (*Figure 1G*; % of co-labeled/total labeled: BM: 2.28 ± 0.46%, EE: 6.8 ± 0.97% p=0.0003 by nested t-test). The results suggest that a cohort of neurons, tagged following EE, reactivate when reintroduced to the same environment, demonstrating memory-specific activation. Therefore, in subsequent experiments, we presumed that cells labeled by task-related c-*Fos*-driven reporter expression represent engram cells. Since EE resulted in greater overall DG neuron labeling and stable reactivation of a subset of neurons after 1 week, we adopted EE as the preferred paradigm to label task-related neuronal ensembles for circuit-level analysis.

We examined tagged neurons in sections from mice that underwent BM navigation and EE exposure to determine the proportional recruitment of SGCs. SGCs were distinguished from GCs by a trained investigator based on (1) the presence of multiple primary dendrites, (2) greater soma width than height, (3) wide dendritic arbor, and/or (4) location in or close to the inner molecular layer (*Figure 1H*). These criteria were based on our prior studies in which unbiased cluster analysis of GC and SGC morphometric data identified the number of dendrites, soma aspect ratio, and dendritic arbor width as the main factors distinguishing the cell types (*Gupta et al., 2020*; *Afrasiabi et al., 2022*). The morphology-based classification revealed that 33.86 ± 2.18% of neurons labeled during BM acquisition and 27.83 ± 1.33% during EE exposure were SGCs (*Figure 1I*; p=0.1143 by nested t-test, based on 40 sections from 4 mice). Since SGCs represent less than 5% of DG projection neurons (*Save et al., 2019*), these data suggest preferential activation of SGCs during dentate-dependent contextual memory formation. Notably, the proportional recruitment of SGCs labeled following behavior was not different between the BM navigation and EE exposure (*Figure 1I*). These findings make a compelling case for leveraging EE exposure to study SGC involvement in dentate-dependent microcircuits.

## Contribution of intrinsic physiology to activity-dependent neuronal labeling

To test if the intrinsic physiology of GCs and SGCs labeled during EE differs from their unlabeled counterparts, we performed whole-cell recordings from labeled- and unlabeled-GCs and SGCs in slices from TRAP2ChR2/eYFP mice 1 week after tamoxifen induction during EE exposure. Labeled and unlabeled neurons in the GC layer and inner molecular layer were visualized under epifluorescence ($\lambda$ =505 nm) and IR/DIC, respectively. Recorded neurons were classified as GC or SGC based on morphology of biocytin-filled neurons (*Figure 2A and B*; *Williams et al., 2007*; *Gupta et al., 2020*; *Afrasiabi et al., 2022*). Depolarizing response to blue light activation (0.9 mW, $\lambda$ =470 nm, 10 ms) of ChR2 was used to functionally validate cell labeling (*Figures 3E and 4E*). Consistent with earlier studies (*Afrasiabi et al., 2022*), there was no cell-type-specific difference in resting membrane potential (RMP) between GCs and SGCs (*Figure 2C*). RMP was also not different between labeled and unlabeled cells within each cell type. Similarly, while the input resistance ($R_{in}$) of RGC was lower than that of GCs, as reported previously (*Williams et al., 2007*; *Afrasiabi et al., 2022*), $R_{in}$ of labeled and unlabeled neurons was not different in either cell type (*Figure 2D*). Examination of responses to a graded current injection revealed divergence of the firing frequency between GCs and SGCs at current injections >400 pA, with GCs showing progressive reduction in frequency with increasing current injection (*Figure 2E–G*) due to an apparent depolarization block. Consistently, the firing frequency in response to +520 pA current was greater in SGCs than that in GCs (*Figure 2H*). Again, these cell-type-specific differences were maintained in both labeled and unlabeled neurons. The action potential (AP) parameters, including threshold, amplitude, half-width, fast afterhyperpolarization (fAHP),

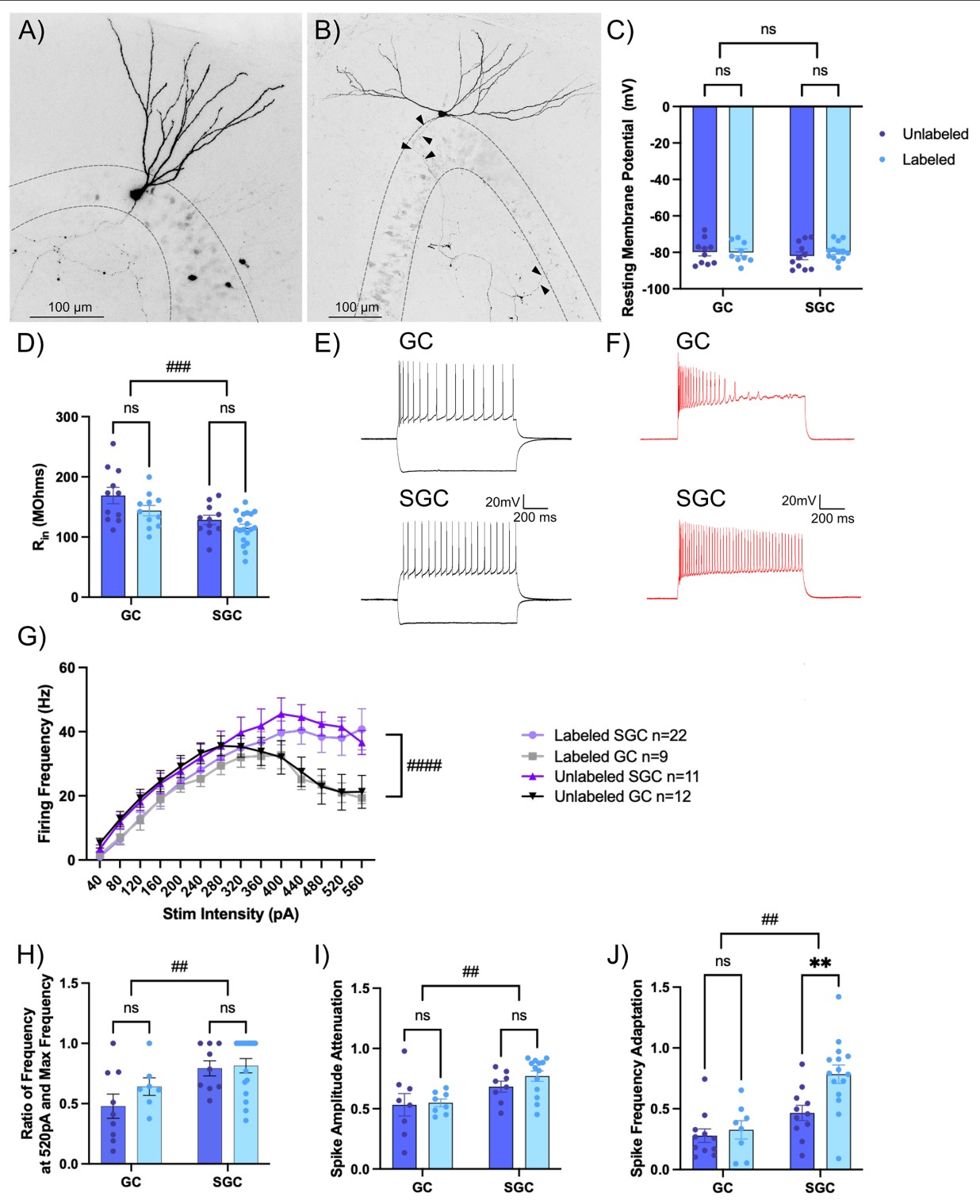

**Figure 2.** Intrinsic differences in frequency adaptation distinguish labeled semilunar granule cells (SGCs). (**A–B**) Representative images of a biocytin-filled granule cell (GC) (**A**) with a narrow dendritic arbor and a smaller somatic width and an SGC (**B**) with wide dendritic span, greater somatic width than height, and axonal projections throughout the molecular and granule cell layer (arrowheads). Maximum intensity projections of confocal image stacks are presented as gray scale, inverted images. (**C–D**) Summary plots of resting membrane potential (RMP in C) and input resistance (R$_{in}$ in D) between labeled and unlabeled GCs and SGCs. # indicates $p<0.05$ for main factor cell type by two-way ANOVA and * indicates $p<0.05$ for labeled versus unlabeled within cell type by Šídák's multiple comparisons post hoc test in n=11–19 cells/group. (**E–F**) Representative cell membrane voltage traces in response to +120 and –200 pA current injections (**E**) and +400 pA current injection (**F**) in GC (top) and SGC (bottom). (**G**) Summary plot of firing

*Figure 2 continued on next page*

*Figure 2 continued*

frequency in response to increasing current injections in labeled and unlabeled SGCs and GCs. #### indicates p<0.0001 for main factor cell type by three-way ANOVA, n=9–22 cells/group. (**H–J**) Summary plots of firing frequency at 520 pA compared to max frequency (**H**), spike amplitude attenuation calculated as ratio between the amplitude of the 15th spike and 1st spike at a current injection of 400 pA (**I**) and spike frequency adaptation (**J**). # indicates p<0.05, ##p<0.01 for main factor cell type by two-way ANOVA and ** indicates p<0.01 for labeled versus unlabeled within cell type by Šídák's multiple comparisons post hoc test in n=8–19 cells/group.

The online version of this article includes the following source data and figure supplement(s) for figure 2:

**Source data 1.** Data underlying *Figure 2C, D, G, H, I and J*.

**Figure supplement 1.** Active properties of labeled and unlabeled granule cells (GCs) and semilunar granule cells (SGCs).

**Figure supplement 1—source data 1.** Data underlying *Figure 2—figure supplement 1A-G*.

medium afterhyperpolarization (mAHP), and latency to first AP, were not different between cell types or labeling of neurons (*Figure 2—figure supplement 1*). Interestingly, GCs showed greater amplitude attenuation during continuous firing, which was not observed in SGCs (*Figure 2I*). Once again, these cell-type-specific differences were retained in both labeled and unlabeled neurons. Finally, SGCs show higher adaptation ratios (ratio of duration between first two and last two spikes in response to 120 pA current injection), indicating less spike frequency adaptation than in GCs, consistent with previous findings (*Williams et al., 2007*). Notably, labeled SGCs showed significantly lower adaptation in firing rate than unlabeled SGCs (*Figure 2J*; GC$_{Labeled}$: 0.33±0.075; GC$_{Unlabeled}$: 0.28±0.056; SGC$_{Labeled}$: 0.78±0.076; SGC$_{Unlabeled}$: 0.47±0.06; two-way RM ANOVA main effect of cell type, p=0.006). In contrast, labeled and unlabeled GCs did not differ in adaptation ratio, indicating that the ability to sustain firing may distinguish labeled SGCs. These data support a role for SGC intrinsic physiology, specifically non-attenuating, less adapting, and persistent firing in their preferential labeling during activity-dependent neuronal tagging.

## Lack of evidence of local feedforward or recurrent excitation between activity-driven neuronal ensembles

Unlike GCs, SGCs have axon collaterals in the inner molecular layer and GC layer (*Figure 2A and B*; *Williams et al., 2007*; *Save et al., 2019*), raising the possibility that they could activate GC dendrites. To test the local feedforward/recurrent excitation hypothesis, we conducted dual patch recordings from labeled neuron pairs to identify potential glutamatergic interconnections (*Figure 3A and C*). Care was taken to ensure that neurons at a depth of 50 μm or more from the surface with visible axons were targeted in order to maximize probability of connections. As noted in *Figure 3B*, a majority of the 32 simultaneously recorded neurons examined between labeled neurons were between SGCs (*Figure 3B*). However, all other possibilities, including GC$_{Labeled}$ to GC$_{Labeled}$, SGC$_{Labeled}$ to GC$_{Labeled}$, and GC$_{Labeled}$ to SGC$_{Labeled}$, were also evaluated. The presence of spontaneous excitatory postsynaptic currents (sEPSCs) in the neurons recorded under voltage clamp served as confirmation of overall circuit and slice health (*Figure 3D*). Additionally, optically evoked (0.9 mW, $\lambda$ =470 nm, 10 Hz, 10 ms pulses) inward currents provided functional validation of reporter expression (*Figure 3E*). Labeled neuronal pairs were tested for glutamatergic synaptic connections by depolarizing one of the neurons in current clamp (400 pA, 10 pulses for 10 ms at 50 Hz) and recording evoked current responses in the other neuron held at –70 mV (*Figure 3F*). The recording configuration was reversed to check for connections in both directions (*Figure 3G*). Despite the presence of sEPSCs, none of the 32 labeled neuronal pairs tested, including SGC$_{Labeled}$ to GC$_{Labeled}$ (n=7) and SGC$_{Labeled}$ to SGC$_{Labeled}$ (n=16), showed functional glutamatergic synaptic connections (*Figure 3F and G*). Although our data do not eliminate the possibility of direct excitatory connections between labeled neurons, they indicate that glutamatergic interconnections are not critical for activation of DG neuronal ensembles.

## Limited evidence for neurons in activity-driven ensembles supporting lateral inhibition

The role of surround inhibition and winner-take-all activation has been proposed as a promising mechanism for establishing memory engrams and mediating dentate processing (*Espinoza et al., 2018*; *Guzman et al., 2021*). SGCs, with preferential recruitment in DG engrams, sustained firing, and hilar axon collaterals, are ideally suited to drive robust feedback inhibition (*Larimer and Strowbridge*,

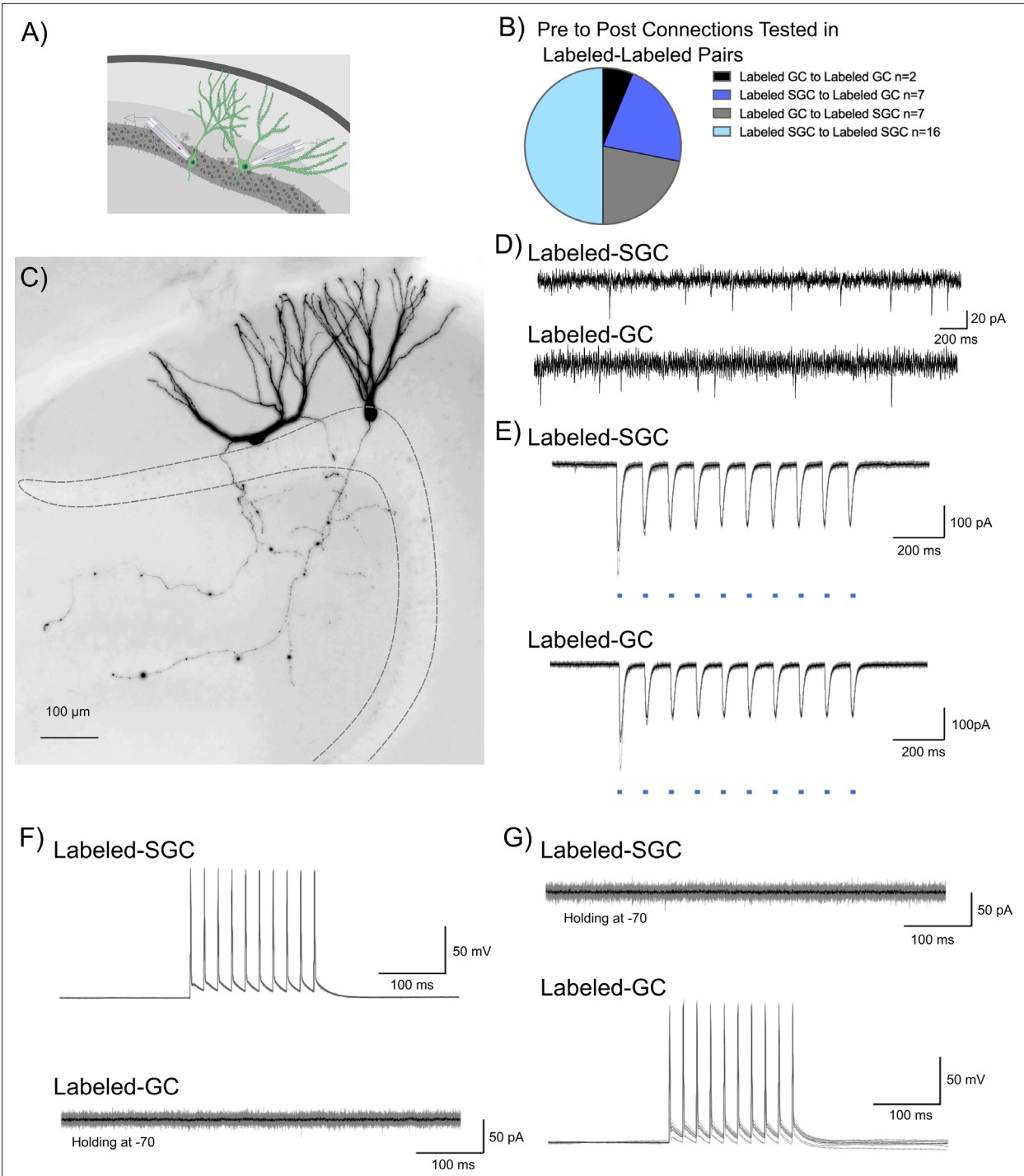

**Figure 3.** Tagged dentate gyrus (DG) neurons do not support mutual excitatory drive. (**A**) Schematic showing dual patch clamp recording from labeled (green) granule cell (GC)-semilunar granule cell (SGC) pair. Created in BioRender.com. (**B**) Summary breakdown of cell-type-specific connections tested in dual recordings from labeled neurons. (**C**) Representative maximum intensity projection of a confocal image stack of a pair of biocytin-filled SGC (left) and GC (right). Images are grayscale and inverted and are overexposed to emphasize the intact axonal arbors in the recorded pair. (**D**) Presence of spontaneous excitatory postsynaptic currents (EPSCs) in the SGC-GC pair in E–G to verify the presence of excitatory inputs and a healthy circuit. (**E**) Light-evoked inward currents validate expression of ChR2 in labeled cell pair. (**F**) Representative traces from a labeled SGC and labeled GC show that depolarization-induced firing in SGC (top) failed to evoke EPSCs in a GC (bottom) recorded in voltage clamp. Individual traces are in gray with average trace overlaid in black. (**G**) Depolarization-induced firing in GC (bottom) fails to evoke EPSCs in an SGC recorded in voltage clamp (top).

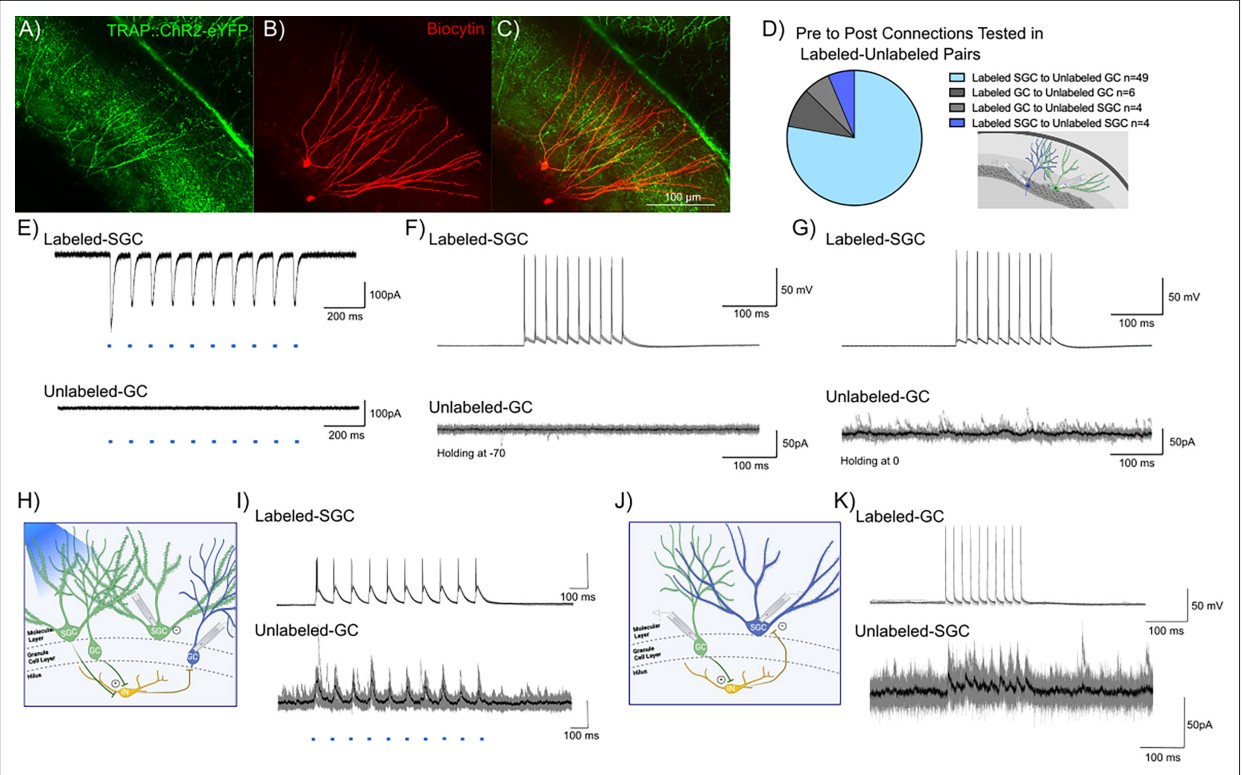

**Figure 4.** Evidence for dentate gyrus (DG) engram neurons supporting sparse feedback inhibition onto non-engram neurons. (**A–C**) Representative confocal image of eYFP-labeled neurons in a TRAP-ChR2-eYFP mouse (**A**) shows biocytin staining (**B**) in a pair of recorded labeled-semilunar granule cell (SGC) and unlabeled-granule cell (GC). Note co-labeling for eYFP and biocytin in the SGC, while the GC does not colocalize eYFP (**C**). (**D**) Summary of cell-type-specific connections tested in dual recordings from labeled and unlabeled neurons. Inset depicts a schematic showing dual patch clamp recording from a labeled (green) SGC and an unlabeled (blue) GC pair. Created with BioRender.com. (**E**) Light-evoked currents validate the expression of ChR2 in the labeled-SGC and lack of response in the unlabeled-GC. (**F–G**) Representative traces from a labeled-SGC and an unlabeled-GC show that depolarization-induced firing in the labeled-SGC (top) failed to evoke excitatory postsynaptic currents (EPSCs) (**F**) and inhibitory postsynaptic currents (IPSCs) (**G**) in the unlabeled-GC. (**H**) Schematic of recording configuration illustrated wide-field optical illumination with labeled neurons (green), unlabeled neurons (blue), and local circuit interneuron (yellow). (**I**) Example traces from a recording in which wide-field optical stimulation evoked inhibitory responses in the unlabeled-GC and firing in the labeled-SGC. Note that the SGC firing by depolarization in the absence of light failed to elicit IPSCs in the same GC. (**J–K**) Schematic with labeled-GC (green), unlabeled-SGC (blue), and local circuit interneuron (yellow) (**J**) and traces from a recorded pair where depolarization of a labeled-GC elicited inhibitory responses in an unlabeled-SGC (**K**). Panels H and J were created with BioRender.com.

The online version of this article includes the following source data and figure supplement(s) for figure 4:

**Figure supplement 1.** Robust feedback inhibition in response to focal activation of a random cohort of granule cells.

**Figure supplement 1—source data 1.** IPSC ampltide data used to generate plots in *Figure 4—figure supplement 1B*.

2010; *Walker et al., 2010*; *Afrasiabi et al., 2022*). We examined the possibility that labeled neurons, particularly labeled SGCs, refine GC activity by mediating feedback surround inhibition of unlabeled GCs. We tested this by performing dual recordings from labeled-unlabeled (L-U) neuronal pairs (*Figure 4A–C*). While the majority of our recordings focused on SGC_Labeled to GC_Unlabeled (49/63 pairs), we also tested for connections between SGC_Labeled to SGC_Unlabeled, GC_Labeled to SGC_Unlabeled, and GC_Labeled to GC_Unlabeled (*Figure 4D*). The ability of wide-field optogenetic activation (0.9 mW, $\lambda$ =470 nm, 10 pulses, for 10 ms at 10 Hz train) to evoke inward currents validated ChR2 expression in labeled neurons. As expected, wide-field blue light stimulation failed to evoke inward currents in unlabeled neurons, confirming the lack of ChR2 expression (*Figure 4E*). The recordings also allowed us to test the unlikely possibility that activation of ChR2-expressing labeled neurons evoked synaptic excitation in unlabeled neurons. Wide-field light activation, which likely activates multiple labeled neurons and axon terminals throughout the slice preparation, did not evoke putative polysynaptic EPSCs in unlabeled neurons (n=63 pairs tested). Consistently, in paired recordings between labeled and unlabeled

neurons, current-evoked firing in labeled cells failed to evoke EPSCs in unlabeled cells (*Figure 4F*), underscoring the lack of glutamatergic connectivity between SGCs and GCs.

In paired recordings from labeled SGCs and unlabeled GCs (n=49), depolarization evoked firing in SGCs (400 pA, 10 ms, 10 pulses, 50 Hz) failed to evoke polysynaptic inhibitory postsynaptic currents (IPSCs) in unlabeled GCs (*Figure 4G*). Notably, despite the lack of evoked IPSCs, the unlabeled GCs received spontaneous IPSCs, indicating that cell and circuit health were not compromised (*Figure 4G*). Interestingly, in 1/49 recordings from $SGC_{Labeled}$ to $GC_{Unlabeled}$ pairs, wide-field optogenetic activation at a light intensity that evoked firing in the recorded labeled SGC evoked IPSCs in the unlabeled GC in the absence of direct synaptic connection between the pairs (*Figure 4H and I*). However, in a majority of trials, wide-field optogenetic activation of both labeled GCs and labeled SGCs failed to evoke IPSCs in unlabeled GCs. Activating labeled neurons did not lead to IPSCs in unlabeled neurons in any of the $GC_{Labeled}$ to $GC_{Unlabeled}$ or $SGC_{Labeled}$ to $SGC_{Unlabeled}$ pairs tested. Unexpectedly, we identified one pair (out of 63) in which current-induced firing in a labeled GC resulted in robust feedback IPSCs in an unlabeled SGC (*Figure 4J and K*). These data identify that labeled GCs can support feedback inhibition of SGCs.

In light of the unexpected paucity in lateral inhibition by engram neurons, we examined whether the circuit connectivity needed to support lateral inhibition is preserved in the slices from mice in which a random cohort of GCs was labeled by transfection with AAV5-CAMKIIa-hChR2(H134R)-eYFP to express ChR2 in excitatory neurons. Using a spatial illumination approach, we optically activated GC somata in three different decreasing circular regions of interest (ROIs) and recorded IPSCs in unlabeled GCs outside the stimulation zone. Focal optical activation of GCs consistently resulted in robust IPSCs in the recorded unlabeled GC with IPSC amplitude decreasing progressively with the size of the ROI (*Figure 4—figure supplement 1*, p<0.05 by one-way ANOVA, from 5 to 8 cells from 3 mice). Collectively, although we find that activation of a focal cohort of neurons supports GC lateral inhibition, our data indicate limited evidence for robust lateral inhibition by neurons labeled during EE exposure onto surrounding unlabeled neurons.

## Labeled neuron pairs receive more correlated spontaneous excitatory inputs

Since microcircuit connectivity and intrinsic physiology could not fully account for task-related coactivation of neurons, we tested the hypothesis that correlated inputs contribute to ensemble activation. First, we evaluated the contribution of AP-driven events to GC sEPSCs. Although the frequency of sEPSC in GCs is low, previous studies have identified a substantial contribution of AP-driven events to GC sEPSCs in slices from rat (*Pernía-Andrade et al., 2012*). In GCs recorded under our experimental conditions in mice, the sodium channel blocker tetrodotoxin (TTX) (1 μM) consistently increased EPSC inter-event interval (IEI) (*Figure 5—figure supplement 1*) by twofold (2.25±0.3-fold increase in IEI from 5.29±0.84 s in aCSF to 12.49±2.84 s in TTX, n=12 cells/3 mice, p=0.0005 by paired t-test). These data identify that approximately half the sEPSCs recorded in GCs represent AP-dependent events and justify analysis of sEPSC in individual neurons and their correlations in neuronal pairs.

To assess whether labeled and unlabeled neurons receive differential glutamatergic drive, we recorded sEPSCs from labeled and unlabeled GCs and SGCs in slices from TRAP2-tdT mice 1 week after tamoxifen induction following EE (*Figure 5A and B*). Recordings from GCs identified that sEPSCs in labeled cells had shorter IEI than unlabeled GCs, indicating more frequent sEPSCs (*Figure 5C and D*, IEI in s, unlabeled: 5.87 (9.94), n=5 cells from 5 mice; labeled: 2.14 (3.71), n=5 cells from 4 mice, p<0.0001 by Kolmogorov-Smirnov [K-S] test, Cohen's d=0.73). Additionally, sEPSC amplitude was also higher in labeled GCs than in unlabeled GCs (*Figure 5D*, amplitude in pA, unlabeled: 16.69 (8.82), n=5 cells from 4 mice; labeled: 20.36 (8.78), n=5 cells from 4 mice, p<0.0001 by K-S test, Cohen's d=0.57). As with GCs, labeled SGCs also had lower sEPSC IEI, indicating higher frequency (*Figure 5E and F*, IEI in s, unlabeled: 2.0 (3.1), n=6 cells from 4 mice; labeled: 1.30 (2.64), n=6 cells from 4 mice, p<0.0001 by K-S test, Cohen's d=0.33) and larger sEPSC amplitude (*Figure 5F*, amplitude in pA, unlabeled: 15.0 (7.08), n=6 cells from 4 mice; labeled: 19.36 (8.64), n=6 cells from 4 mice, p<0.0001 by K-S test, Cohen's d=0.58) than in unlabeled SGCs. Thus, a greater input drive rather than local circuit connectivity distinguishes labeled GCs and SGCs from their unlabeled counterparts.

To evaluate whether temporally correlated inputs contribute to ensemble labeling during EE exposure, we analyzed sEPSC in dual recordings from labeled-labeled (L-L) and L-U pairs for temporal

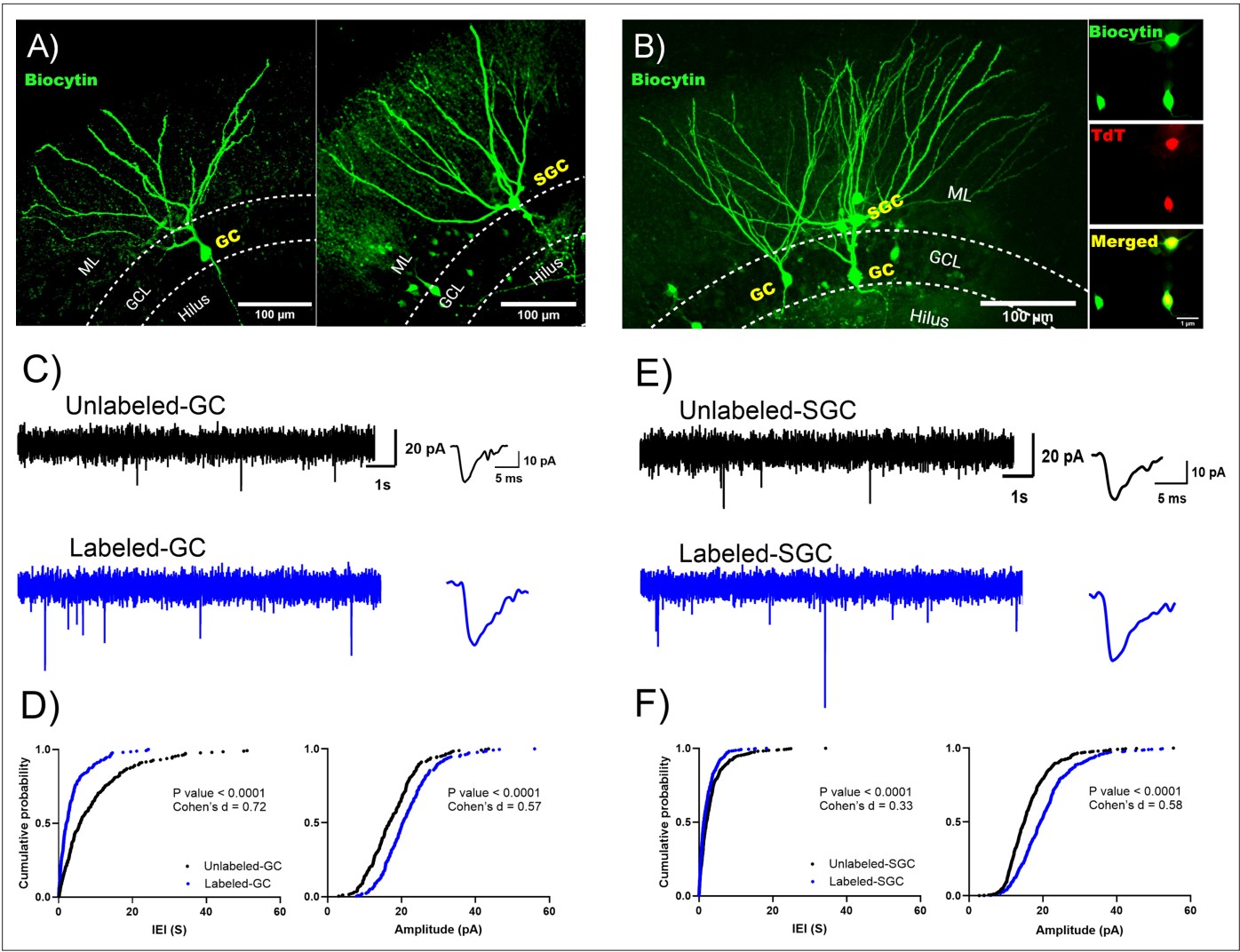

**Figure 5.** Labeled granule cells (GCs) and semilunar granule cells (SGCs) receive more frequent spontaneous excitatory inputs than unlabeled cells. (**A–B**) Representative images of a biocytin-filled unlabeled GC (left panel) and SGC (right panel) (**A**) and image of a slice in which an unlabeled GC was recorded alongside a labeled GC and SGC. (**B**) Inset in B shows biocytin fill, tdT labeling, and merge of the somata to illustrate co-labeling. (**C**) Representative current traces illustrate spontaneous excitatory postsynaptic currents (sEPSCs) in an unlabeled (top) and labeled (bottom) GC. Panels to the right: Representative average sEPSCs trace. (**D**) Cumulative probability plot of sEPSC inter-event interval (left panel) and amplitude (right panel) in labeled (black) and unlabeled (blue) GC. (**E**) Representative current traces illustrate sEPSCs in an unlabeled (top) and labeled (bottom) SGC. Panels to the right: Representative average sEPSCs trace. (**F**) Cumulative probability plot of sEPSC inter-event interval (left panel) and amplitude (right panel) in labeled (black) and unlabeled (blue) SGC. p-Value by Kolmogorov-Smirnov test is indicated in the figure, n=5–6 cells/group. Effect size estimate using Cohen's d is indicated in the plots.

The online version of this article includes the following source data and figure supplement(s) for figure 5:

**Source data 1.** sIPSC interevent interval and amplitude data used to generate *Figure 5D and F*.

**Figure supplement 1.** Spontaneous excitatory postsynaptic currents (EPSCs) in dentate granule cells (GCs) include action potential-driven events.

**Figure supplement 1—source data 1.** IPSC interevent interval data used to generate *Figure 5—figure supplement 1B*.

correlation of synaptic event times (*Figure 6A–C*). Note that since the intent was to determine the input correlation depending on labeling status of the cell pairs rather than based on cell type, we combined datasets for pairs that included GCs and/or SGCs. To assess sEPSC temporal correlation, we defined *peri-occurrence* as the maximum cross-correlation of sEPSCs in the two recorded neurons within a *detection window* and *co-occurrence* as the cross-correlation of sEPSCs within a more restricted time *bin duration* centered around 0 ms (*Figure 6D*). To avoid potential for specious correlations due to differences in event frequency, we sub-selected cell pairs in which the recording durations, event

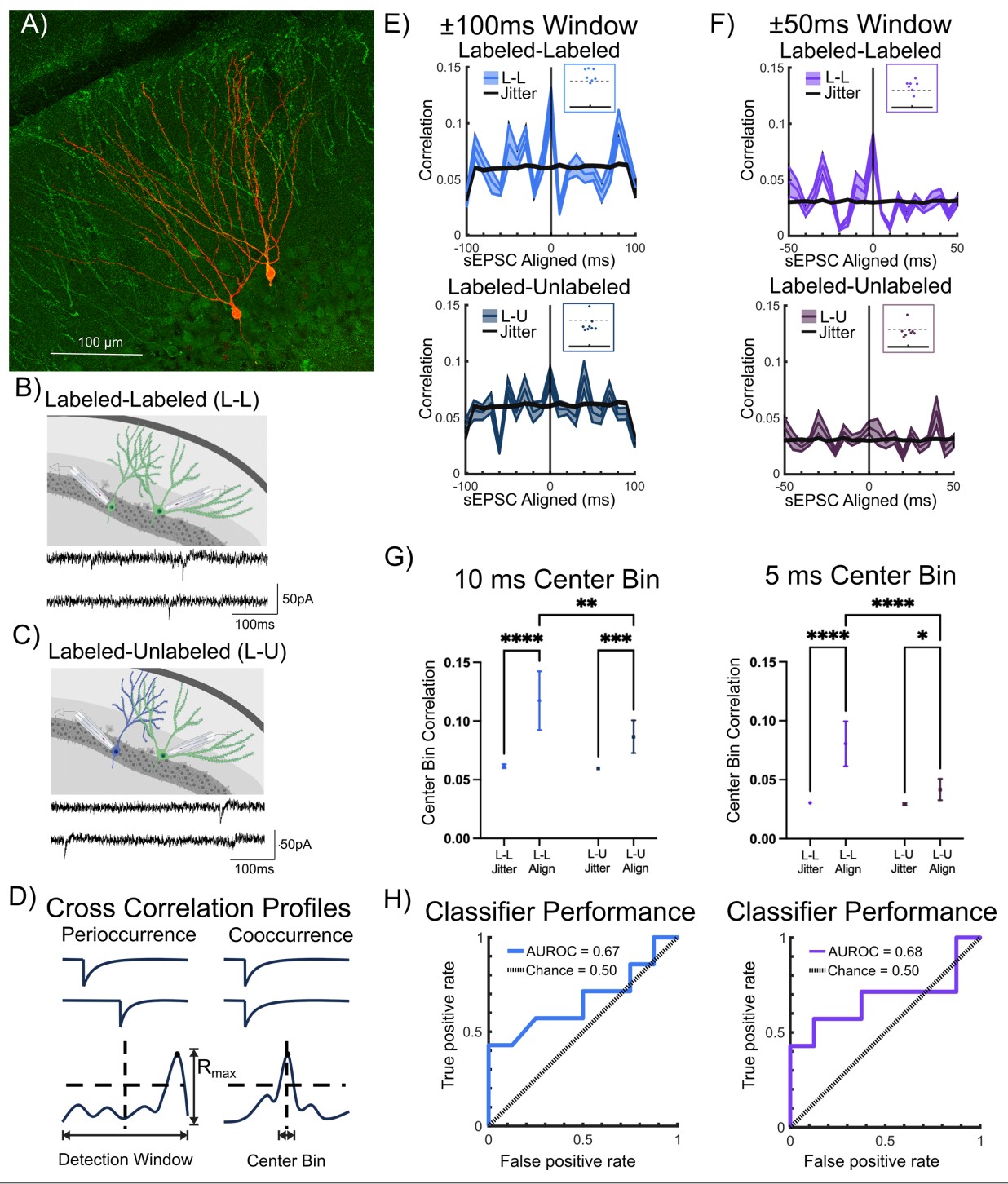

**Figure 6.** Correlated spontaneous excitatory inputs to labeled pairs. (**A**) Representative confocal image of eYFP-labeled and biocytin-stained neurons in a TRAP-ChR2-eYFP mouse. (**B**) Schematic for labeled-labeled (L–L) dual recordings with representative example of spontaneous excitatory postsynaptic currents (sEPSCs) in an L-L pair below. (**C**) Schematic for labeled-unlabeled (L–U) dual recordings with representative example of sEPSCs in an L-U pair below. (**D**) Schematic for session-wise cross-correlation profiles (CCPs) defined by correlations exceeding a 2 standard deviation (SD) threshold above the total mean correlation: EPSC peri-occurrence was tested as event time CCP exceeding threshold within full detection window; co-occurrence was defined as event time CCP exceeding threshold within center bin of detection window. (**E**) CCP from recordings from L-L pairs analyzed with ±100 ms detection window (bright blue, n=7). Overlaid jittered data (black) was developed by appending the event timing of one cell with a randomized lead/ lag of ±0.5 s for 100 iterations (top panel). Inset: Plot of maximum correlations (R$_{max}$) in relation to the dashed line representing 2×SD = 0.15. CCP in recordings from L-U pairs analyzed with ±100 ms detection window (dark blue, n=8). Corresponding jittered data, developed as detailed above, is

*Figure 6 continued on next page*

*Figure 6 continued*

overlaid in black (bottom panel). Inset: Plot of $R_{max}$ in relation to the dashed line representing 2×SD = 0.15. (**F**) CCP from sessions with recordings from L-L pairs analyzed with ±50 ms detection window from L-L pairs (bright purple, n=7) with jittered data developed as detailed above is overlaid in black (top panel). Inset: Plot of $R_{max}$ in relation to the dashed line representing 2×SD = 0.10. CCP from recordings in L-U pairs analyzed with ±50 ms detection window (dark purple, n=8) with corresponding jittered data overlaid in black (bottom panel). Inset: $R_{max}$ in relation to the dashed line representing 2×SD = 0.10. (**G**) Comparison of center bin correlation between L-L versus L-U pairs in aligned (align-recorded) versus jittered (Jitter-simulated) data, analyzed using ±100 ms detection window (left, colors as in E) and using ±50 ms detection window (right, colors as in F). (**H**) Center bin classifier performance (solid line) compared to chance performance (dashed line, colors as in E and F, respectively) plotted as area under the receiver operating characteristic (ROC) curve (AUROC) between L-L (true positive rate) and L-U (false positive rate) for analysis using ±100 ms detection window (left panel) and for analysis using ±50 ms detection window (right panel). Data presented as mean ± SEM (dual recording sessions), * indicate p<0.05, ** indicates p<0.01; *** indicates p<0.001, **** indicates p<0.0001; two-way ANOVA with Šídák's multiple comparisons post hoc tests. Panels B and C were created with BioRender.com.

The online version of this article includes the following source data and figure supplement(s) for figure 6:

**Source data 1.** Data used to generate *Figure 6E, F and H*.

**Figure supplement 1.** Labeled-labeled (L-L) and labeled-unlabeled (L-U) sessions do not differ in event rates.

**Figure supplement 1—source data 1.** Raw data for plots in *Figure 6—figure supplement 1B and C*.

**Figure supplement 2.** Example spontaneous excitatory postsynaptic current (sEPSC) cross-correlation profiles (CCPs).

**Figure supplement 2—source data 1.** Raw data used to generate plots in *Figure 6—figure supplement 2*.

count, and activity rates were not different between L-L and L-U pairs (*Figure 6—figure supplement 1*, number of sEPSCs in counts, L-L: 439±52.83, L-U: 412±48.77, p=0.71; recording duration in s, L-L: 498±43.76, L-U: 397±48.0, p=0.15; spike rate in Hz, L-L: 0.89±0.09, L-U: 1.06±0.11, p=0.23).

We selected detection windows of ±100 ms and ±50 ms with 10 ms and 5 ms bin width, respectively, to develop cross-correlation profiles (CCPs) of L-L and L-U sEPSC event times as detailed in the Materials and methods. Example recording sessions with CCP for co-occurrence in L-L pairs, no coincidence in L-U pairs, and peri-occurrence in L-L pairs, analyzed using a ±100 ms detection window are illustrated in *Figure 6—figure supplement 2*. A predetermined threshold of 2 standard deviations (2SD) above the mean correlation was adopted to assess potential differences in temporal correlation between L-L and L-U pairs. Peri-occurrence, quantified as the maximum cross-correlation in the detection window ($R_{max}$), was significantly higher in L-L than in L-U pairs for the ±100 ms detection window (*Figure 6*, $R_{max}$ in ±100 ms window, L-L: 0.184±0.014, L-U: 0.126±0.016, p=0.018 by t-test in 7 L-L and 8 L-U pairs). While peri-occurrence in the ±50 ms detection window trended higher in L-L pairs, this was not significant (*Figure 6*, $R_{max}$ in ±50 ms window, L-L: 0.124±0.013, L-U: 0.093±0.014, p=0.12 by t-test in 7 L-L and 8 L-U pairs). Notably, the $R_{max}$ in sEPSC event times from 6/7 L-L pairs crossed the threshold within each detection window and failed to do so in 7/8 recordings in L-U pairs. These findings were consistent regardless of whether we adopted ±100 ms or ±50 ms detection windows (*Figure 6E and F*, insets).

To determine whether event correlations in neuronal pairs deviated from randomness, event time correlations in the recorded (*temporally aligned*) data were compared with the correlations developed from corresponding temporally *jittered* datasets (*Figure 6E and F*). Event time center bin correlations in the recorded (*temporally aligned*) datasets were significantly higher than in the *jittered* datasets for both detection windows (*Figure 6G* main effect of data alignment, 10 ms bin: $F_{(1,211)}$=63.16, p<0.0001 by two-way ANOVA, 5 ms bin, $F_{(1,211)}$=71.60, p<0.0001 by two-way ANOVA). Co-occurrence, quantified as the correlation in the center bin, was higher in L-L pairs both within the 10 ms bin (*Figure 6E and G*, 10 ms center bin correlation, L-L: 0.117±0.025, L-U: 0.087±0.014, p=0.0075 by two-way ANOVA in 7 L-L and 8 L-U pairs) and within the 5 ms bin (*Figure 6F and G*, 5 ms center bin correlation, L-L: 0.081±0.019, L-U: 0.042±0.009, p<0.0001 by two-way ANOVA in 7 L-L and 8 L-U pairs).

Finally, we evaluated the ability of co-occurring events to predict an L-L versus L-U recording session. The receiver operating characteristic (ROC) curve of true versus false positive rates defined the area under the ROC curve (AUROC) for the center 10 ms and 5 ms bin across the ±100 ms and ±50 ms detection windows, respectively (*Figure 6H*). Center bin classification performed better than chance at predicting whether a recorded session was L-L versus L-U ($AUROC_{Chance}$ = 50%; $AUROC_{10msCenter}$ = 66.96%; $AUROC_{5msCenter}$ = 71.43%). Thus, the 10 ms and 5 ms center event correlations were predictive of whether a pair of recorded neurons was likely to be a pair of labeled neurons or a L-U

pair. Together, these results support a role for correlated inputs in driving shared neuronal activation during contextual memory formation.

## Discussion

The recent characterization of SGCs as a unique dentate projection neuron subtype overrepresented among behaviorally recruited DG neurons has raised the intriguing possibility that SGCs may play a distinct role in shaping DG ensemble activity (**Walker et al., 2010**; **Erwin et al., 2020**; **Afrasiabi et al., 2022**). Here, we evaluated competing hypotheses involving mechanisms governing recruitment of GC and SGC populations during a behavioral experience. Our data identify that *intrinsic properties of SGCs,* specifically their *less adapting firing* characteristics, likely enable preferential recruitment of SGCs among neurons labeled based on IEG expression. At the circuit level, neurons activated during a behavioral experience received more frequent and larger excitatory synaptic input than those not engaged in the task. Notably, neurons in a shared DG ensemble receive *more correlated spontaneous excitatory inputs* than neurons without shared activation, suggesting a role for common afferent inputs in behaviorally driven ensemble recruitment. Whereas GCs and downstream CA3 neurons with shared behavioral activation develop preferential glutamatergic connectivity (**Ryan et al., 2015**), we found no evidence for local feedforward or recurrent excitation in DG neurons labeled as part of a memory trace. Unexpectedly, although lateral inhibition has been proposed as a mechanism for dentate engram refinement (**Stefanelli et al., 2016**), our experiments revealed that activation of labeled DG neurons rarely drove inhibitory synaptic currents in unlabeled neurons. Interestingly, we find approximately a third of the projection neurons activated as a part of a dentate-dependent spatial navigation task and during exposure to EE had morphology consistent with SGCs. It is possible that the reduced AP accommodation and attenuation in SGCs contributes to enhanced SGC firing and c-*Fos* expression during afferent activation. This greater activity-dependent c-*Fos* expression in SGCs may result in their preferential labeling as part of neuronal ensembles activated during behavioral tasks. Together, these results identify that behaviorally relevant activation of SGCs and GCs in DG neuronal ensembles is determined by a combination of sustained firing characteristics of SGCs, enhanced glutamatergic inputs, and shared afferent drive rather than by selective circuit-level refinement by recurrent excitation or by lateral inhibition.

Studies evaluating mechanisms of ensemble recruitment during fear and aversive memory encoding, by experimentally enhancing excitability or CREB expression in a sparse population of amygdala neurons, have proposed that neurons with higher excitability outcompete neighboring cells for allocation to behaviorally activated neuronal ensembles (**Han et al., 2007**; **Zhou et al., 2009**; **Sano et al., 2014**; **Yiu et al., 2014**; **Gouty-Colomer et al., 2016**). However, our data revealed no difference in intrinsic physiology and active properties between labeled and unlabeled GCs. This is consistent with a previous report that firing threshold and input resistance of GCs labeled during fear conditioning and recorded 24 hr later were not different from unlabeled GCs (**Ryan et al., 2015**), although GCs and SGCs were not explicitly distinguished. In addition to previously reported less adapting firing in SGCs than in GCs (**Walker et al., 2010**; **Afrasiabi et al., 2022**), we find reduced spike amplitude attenuation in SGCs resulting in more sustained firing, particularly in response to large current injections. Moreover, in the first direct comparison of behaviorally recruited and unlabeled SGC, we identify that labeled SGCs had lower spike frequency adaptation than unlabeled SGCs, indicating that sustained firing may predispose SGCs to activation during behavioral encoding. Indeed, the sustained firing during depolarizing current injections larger than 400 pA (this study) and in response to afferent input (**Larimer and Strowbridge, 2010**; **Afrasiabi et al., 2022**) are quintessential functional differences between GCs and SGCs. This sustained SGC firing is ideally suited to induce more IEG (c-*Fos* or ARC) expression and could contribute to higher-than-expected labeling of SGCs during memory engram encoding. This is also supported by the enrichment of activity-dependent markers, including PENK, in behaviorally activated DG neurons, such as SGCs labeled in TRAP2 mice (**Erwin et al., 2020**). Since the c-*Fos*-dependent ensemble labeling approach requires time for reporter expression, our experimental design does not allow a comparison of neuronal excitability at or before task performance. Nevertheless, our data demonstrating more sustained firing in SGCs and selectively reduced adaptation in labeled SGCs supports a role for greater neuronal activity in preferential recruitment of SGCs to task-related dentate engrams.

It is possible that the sustained firing in SGCs, as well as higher NMDAR-mediated currents (*Larimer and Strowbridge, 2010*), contribute to their increased representation (~30%) among behaviorally tagged DG neuronal ensembles compared to their relative population (~3–5%) (*Save et al., 2019*). Furthermore, the wider dendritic arbors of SGCs are ideally positioned to receive distributed inputs and could support their preferential recruitment during behaviors. In this regard, whether SGCs and GCs differ in the inputs they receive from the entorhinal cortex is currently unknown. Although we find higher-than-expected SGCs among neuronal ensembles, the SGC labeling during BM and EE is considerably lower than the approximate 80% representation in DG engrams reported previously following novel environment exposure (*Erwin et al., 2020*). It is possible that restricting analysis to a subset of neurons filled during physiological recordings contributed to overrepresentation of SGCs among engram neurons in prior studies (*Erwin et al., 2020*). This is not surprising because SGCs, especially those in the sparsely populated molecular layer, are more readily visualized and accessed for patch physiology than labeled GCs in the densely packed cell layer. Indeed, SGCs represent greater than 70% of the labeled neurons recorded in our study. Thus, our manual classification of sparsely labeled neurons by an expert investigator using previously validated morphometric features (*Gupta et al., 2020*; *Afrasiabi et al., 2022*) is likely to more accurately reflect the proportional labeling of SGCs.

Neurons in shared ensembles, including the DG to CA3 projections, show preferential connectivity and selective synaptic strengthening (*Ryan et al., 2015*; *Rao-Ruiz et al., 2021*). While connectivity between GCs is rarely observed in the healthy DG, whether SGCs with axon collaterals in the molecular layer make functional synaptic contacts on GC has not been examined. We leveraged findings that DG ensembles stably reactivate and engage downstream circuits up to 12 days after encoding (*Kitamura et al., 2017*) to evaluate local connectivity among SGCs and GCs in behaviorally recruited DG ensembles 1 week after encoding. Our paired recordings did not find evidence for glutamatergic connections between labeled SGCs and GCs, indicating that local DG engram refinement is not supported by mutual synaptic strengthening. Moreover, we observed no evidence of direct synaptic connectivity between GCs and SGCs, regardless of whether the neurons were labeled or unlabeled. While it is possible that slice preparation could sever axon collaterals, we routinely record from cells over 50 µm below the surface and recovered extensive axon collaterals from SGC-GC pairs and excluded cells in which axon collaterals were not visualized. Moreover, SGCs have compact axonal distribution (*Gupta et al., 2020*), minimizing the possibility that lack of connections was a consequence of severed axons. Furthermore, even wide-field illumination to activate ChR2-positive terminals failed to evoke EPSCs in unlabeled GCs or SGCs, consistent with lack of connectivity between labeled and unlabeled SGC-GC pairs. Taken together with the evidence for increased overlap between DG ensembles labeled during encoding and recall, the limited glutamatergic interconnectivity among GCs and SGCs supports an instructive role for afferent inputs in DG ensemble recruitment. Indeed, we found that labeled neurons received more frequent and higher amplitude spontaneous glutamatergic inputs than corresponding unlabeled cells, suggesting that strengthening of shared inputs may contribute to ensemble maintenance. Consistent with the proposal that afferent inputs contribute to DG ensemble recruitment, we found that the event timing of spontaneous glutamatergic inputs to pairs of labeled DG neurons was more correlated than inputs to L-U pairs, suggesting that labeled neuronal pairs may receive correlated input streams. Moreover, input correlation was effective in discriminating between L-L versus L-U neuronal pairs, further supporting the role for input-dependent recruitment of neuronal ensembles.

Lateral inhibition has long been considered a promising mechanism for discriminating among behaviorally relevant DG neuronal ensembles (*Espinoza et al., 2018*; *Cayco-Gajic and Silver, 2019*; *Guzman et al., 2021*; *Borzello et al., 2023*). Although GC-mediated lateral inhibition of adjacent GCs is sparse (*Espinoza et al., 2018*; *Braganza et al., 2020*), focal activation of a random cohort of task-unrelated GCs, labeled with ChR2 by viral transfection, was shown to mediate surround inhibition of GCs (*Stefanelli et al., 2016*; *Braganza et al., 2020*). Moreover, SGCs, with their ability to robustly activate feedback inhibition (*Larimer and Strowbridge, 2010*; *Afrasiabi et al., 2022*), have been proposed as an ideal cell type to drive surround inhibition (*Walker et al., 2010*). However, unlike the findings based on focal activation of random GCs (*Stefanelli et al., 2016*), our analysis of SGCs and GCs tagged during naturalistic behavior found limited evidence for lateral inhibition of GCs. Even wide-field optical stimulation of labeled neurons, which would be expected to activate labeled terminals on interneurons due to the recruitment of intact and severed axons, rarely elicited

lateral inhibition (1 of 55 recordings). Our control experiments demonstrate that, unlike activation of sparse behaviorally labeled neurons, focal optical activation of cohorts of GCs labeled with ChR2 based on CAMKII expression supported robust feedback inhibition (*Figure 4—figure supplement 1*) in our slice preparation. Moreover, we have consistently recorded unitary IPSCs in DG interneuron-interneuron and interneuron-GC pairs (*Yu et al., 2015*; *Yu et al., 2016*; *Proddutur et al., 2023*), demonstrating that the circuit needed to support lateral inhibition is present in our slice preparation. Collectively, our results suggest that the sparse recruitment of behaviorally labeled ensembles may not be sufficient to elicit lateral inhibition. Our results are consistent with prior findings that focal activation of 2–4% of densely packed GCs is needed to recruit lateral inhibition in the DG (*Braganza et al., 2020*). Importantly, we identify that sparse behaviorally tagged ensembles are insufficient to support the kind of lateral inhibition observed during focal activation of high-density virally labeled GCs (*Stefanelli et al., 2016*; *Braganza et al., 2020* and *Figure 4—figure supplement 1*). Additionally, while modulating interneuron activity can constrain engram size by regulating network excitability (*Morrison et al., 2016*; *Stefanelli et al., 2016*), our microcircuit analyses suggest that SGCs have a limited role in ensemble refinement by lateral inhibition. In this context, the possibility that slice recordings lead to underestimation of feedback dendritic inhibition cannot be ruled out. Curiously, we identified inhibition from a labeled GC to an unlabeled SGC and optically induced inhibition of an unlabeled GC, which does indicate the presence of sparse lateral inhibition in the circuit. Overall, while there is considerable evidence for robust lateral inhibition in regulating DG activity, our data do not support the hypothesis that a sparse population of behaviorally active SGCs and GCs support ensemble refinement by surround inhibition.

The non-fear-based contextual behavioral paradigms adopted in this study are known to engage the DG. However, they resulted in considerably sparser labeling than reported in contextual fear conditioning paradigms adopted in prior studies (*Liu et al., 2012*; *Liu et al., 2014*; *Ryan et al., 2015*; *Roy et al., 2017*). In order to specifically target neuronal cohorts activated during DG-dependent spatial learning, we initially examined neuronal activation during the BM spatial navigation task involving spatial learning over multiple trials (*Gawel et al., 2019*). Tamoxifen treatment on day 6 labeled a sparse cohort of neurons, which was not compatible with circuit-level analysis. Since the DG is preferentially activated by novelty (*Hainmueller and Bartos, 2020*; *Mazurkiewicz et al., 2022*; *Borzello et al., 2023*), we reasoned that learning-related decrease in novelty may have contributed to sparse DG labeling during day 6 of BM spatial navigation. Consistent with a role for novelty in DG ensemble activation (*Mazurkiewicz et al., 2022*), tamoxifen induction during a single episode of EE exposure reliably labeled a larger cohort of DG neurons, thereby enabling circuit analysis. Moreover, our demonstration of significantly greater co-labeling of tdT neurons with c-Fos upon a second exposure to the same environment, than following prior exposure to the BM, confirmed task-specific neuronal labeling. While consistent with the rates of reactivation observed in fear conditioning experiments (*DeNardo et al., 2019*), c-*Fos* co-labeling was observed in less than 10% of the tdT-positive neurons after re-exposure to the environment, suggesting that not all tdT-labeled neurons may be behaviorally relevant. A related caveat is the possibility that use of the TRAP2 system may miss active neurons expressing other IEGs (*Heroux et al., 2018*). Nevertheless, c-*Fos*-driven labeling in TRAP2 mice remains the current best approach for activity-dependent labeling, especially of DG neurons (*Kawashima et al., 2014*).

In summary, we find that SGCs represent about a third of dentate projection neurons labeled based on c-*Fos* expression during the contextual memory encoding, which is a considerable overrepresentation relative to their known population density. We propose that their unique sustained firing characteristics and temporal precision of afferent inputs may support their preferential labeling during activity-dependent labeling of memory ensembles. Taken together, these data support a role for correlated inputs, the ability to sustain AP firing, and sparse surround inhibition rather than glutamatergic interconnectivity as key determinants for recruitment of neurons to dentate memory ensembles.

# Materials and methods

## Key resources table

| Reagent type (species) or resource | Designation | Source or reference | Identifiers | Additional information |
|---|---|---|---|---|
| Strain, strain background (*Mus musculus*) | C57BL/6J | The Jackson Laboratory | MSR_JAX: 000664 | Male and female |
| Strain, strain background (*Mus musculus*) | Fos-cre[er] (Fos[tm1.1(Cre/Ert2)Luo]/J) | The Jackson Laboratory | JAX Stock: 030323 | Male and female |
| Strain, strain background (*Mus musculus*) | B6;129S6-Gt(ROSA) 26Sor t m14(CAG-tdTomato) Hze/J | The Jackson Laboratory | JAX Stock: 007908 | Male and female |
| Strain, strain background (*Mus musculus*) | B6;129S-Gt(ROSA)[26Sortm32(CAG]B6;129S-Gt(ROSA)[COP4*H134R/EYFP)Hze/J] | The Jackson Laboratory | JAX Stock: 12659 | Male and female |
| Strain, strain background (*AAV*) | AAV5-CaMKIIa-hChR2(H134A)-EYFP | Addgene | Plasmid # 26969 | |
| Antibody | c-*Fos* (9f6) rabbit antibody (Monoclonal) | Cell Signaling Technology | 2250S /RRID:AB_2247211 | 1:750 |
| Antibody | Goat anti-rabbit Alexa Fluor 488 secondary antibody (Polyclonal) | Abcam | 150077/ RRID:AB_2630356 | 1:500 |
| Antibody | Chicken Anti-Green Fluorescent Protein Antibody (Polyclonal) | Aves Labs | AB_2307313 | 1:500 |
| Antibody | Goat anti-Chicken Alexa Fluor 488 (Polyclonal) | Abcam | 150169/ RRID:AB_2636803 | 1:500 |
| Chemical compound, drug | 4-Hydroxytamoxifen | Sigma | H7904-25MG | |
| Chemical compound, drug | Tetrodotoxin | Tocris | Tetrodotoxin | Male and female |
| Chemical compound, drug | Gabazine-SR95531 | Tocris | SR95531 | Male and female |
| Software, algorithm | Easy Electrophysiology | Easy Electrophysiology Ltd | v2.6.3 | https://www.easyelectrophysiology.com/ |
| Software, algorithm | BUNS analysis software | *Illouz et al., 2016* | http://okunlab.wix.com/okunlab | |
| Software, algorithm | pClamp10-Data Acquisition | Molecular Devices | https://www.moleculardevices.com | |
| Software, algorithm | Anymaze | Stoelting Co. | https://www.any-maze.com/ | |
| Software, algorithm | MATLAB | MathWorks | R2024a | |
| Software, algorithm | Prism 10 | GraphPad | 10.4.1 | |
| Other | Custom code for correlation analysis | *VijiSanthakumarLab, 2025* | https://github.com/VijiSanthakumarLab/eLife_Correlation_Cells_2025 | |
| Other | Alexa-594 streptavidin conjugate | Thermo Fisher | S11227 | 1:1000 |
| Other | Vectashield | Vector Labs | NC9524612 | |
| Other | Goat serum | *Sigma* | SIAL-G6767-100ML | |
| Other | Barnes maze table | Maze Engineers | https://conductscience.com/maze/ | |

## Animals

All experiments were conducted under IACUC protocols approved by the University of California at Riverside and conformed with ARRIVE guidelines. c-*Fos* mice (TRAP2: Fostm2.1[(icre/ERT2)Luo/J]; Jackson Laboratories #030323) were back-crossed with C57BL6/N and were either bred with reporter line tdT-Ai14 mice (B6;129S6-Gt(ROSA)[26Sortm14(CAG-tdTomato)Hze/J]; Jackson Laboratories # 007908) to create TRAP2-tdT mice or reporter line ChR2-YFP (B6;129S-Gt(ROSA)[26Sortm32(CAG-COP4*H134R/EYFP)Hze/J]; Jackson Laboratories #12 569) to create TRAP2-ChR2/eYFP mice. Male and female TRAP2-tdT and TRAP2-ChR2/eYFP mice 4–8 weeks of age were used in experiments. Mice were housed with littermates (up to 5 mice per cage) in a 12/12 hr light/dark cycle. Food and water were provided ad libitum.

## Behavioral training and engram labeling

Male and female experimental mice were trained in a spatial learning BM task or placed in an EE for 3 hr followed by tamoxifen induction to induce Cre recombinase as detailed below. Since we observed

TRAP2 mice exhibiting considerable litter to litter variability in tdT labeling following identical treatments in preliminary studies (not shown), we used littermate pairs for the following studies: *BM*: 4- to 6-week-old male and female TRAP2-tdT mice were trained in a spatial memory task on a BM table (Maze Engineers, https://conductscience.com/maze/), 92 cm in diameter with 20 holes (5 cm diameter each). One hole was equipped with a false floor installed with a removable escape box that could be traded out for an additional false floor piece. The maze was set up in the middle of four curtain walls with two bright lights and a camera for recording above the maze. Different sets of visual cues (various shapes cut from felt) were attached to the curtain for spatial orientation. The escape hole was positioned in between two visual cues. Animals were held in their home cage outside of the curtain in a dark room until their turn to run the trial. We observed that these mice were hyperactive; therefore, mice were housed individually on the day before training (day 0). Mice were habituated to the behavior room in their home cages for at least 1 hr before training on day 1, habituated to the arena by placing them in the starter cup on the table for 1 min, and guided by gently moving the starter cup to the escape box (in a temporary location different from the experimental location). During task acquisition training on days 1–6, mice performed three 180 s trials during which the mouse explored the maze to find the escape box. The three trials were separated by a minimum of 15 min ITIs. If mice failed to locate the escape box at the end of the 180 s, the experimenter guided them to the escape box and then placed them back into their home cage. On day 6 of BM acquisition, mice were brought to the room 5 hr before testing and received 4-hydroxy tamoxifen (4-OHT, 50 mg/kg i.p.) 15 min prior to the first acquisition session. 4-OHT was prepared as described previously (*DeNardo et al., 2019*). Briefly, 4-OHT was dissolved in 100% ethanol at a concentration of 20 mg/mL by sonicating solution at 37°C for 30 min or until dissolved, aliquoted, and stored at –20°C. On the day of injection, 4-OHT was redissolved by sonicating solution at 37°C for 10 min. A 1:4 mixture of castor oil and sunflower seed oil, respectively, was added for a final concentration of 10 mg/mL. The remaining ethanol in solution was evaporated by speed vacuuming in a centrifuge (*DeNardo et al., 2019*). Behavior in the BM paradigm was analyzed using Anymaze software by a blinded experimenter. Additionally, support vector machine-based, automated, BUNS classification algorithm and a nonarbitrary numerical cognitive score based on the BUNS analysis (*Illouz et al., 2016*) were used to evaluate the use of spatial strategy for BM.

### Enriched environment

Experimental TRAP2-tdT and TRAP2-ChR2/eYFP mice were housed in an EE consisting of an oversized cage filled with multiple tunnels, extra nestlets, a metal swing, and a few huts for the animals to interact with for 3 hr. Mice received 4-OHT (50 mg/kg i.p.) 90 min into their 3 hr of enrichment. Animals were left in the room for an additional 5 hr to limit neuronal activity labeling not related to the behavioral paradigm. In a subset of experiments (*Figure 1*), 7 days following 4-OHT induction, littermate cohorts of TRAP2-tdT mice that underwent BM acquisition or EE exposure were placed with their respective pair into the EE for 2 hr and then immediately sacrificed by perfusion with 4% paraformaldehyde (PFA) upon removal from the EE. TRAP2-ChR2/eYFP mice induced after EE exposure were sacrificed a week later for electrophysiology (*Figures 2–6*; and associated figure supplements).

### Immunohistochemistry and cell morphology

TRAP2-tdT mice, 90 min following EE exposure, were transcardially perfused with PBS followed by a 4% PFA while under euthasol anesthesia. The brains were held in the 4% PFA at 4°C for 3 hr before being transferred to PBS. Coronal brain sections (50 μm) were obtained using a Leica vt100s vibratome, and five sequential sections, each 250 μm apart across the septotemporal axis, were immunostained for c-*Fos* and analyzed for quantification. Free-floating sections were blocked in 10% goat serum in PBS with 0.3% Triton X-100 for 1 hr. Sections were incubated in 4°C overnight in primary antibody for c-*Fos* (1:750, Rabbit mAb Cell Signaling Technology, cat #2250). The following day, sections were incubated in goat anti-rabbit Alexa Fluor 488 secondary antibody (1:500 Abcam, cat #150077) for 1 hr.

Slices from TRAP2-ChR2/eYFP mice that were used in electrophysiological studies were fixed in 0.1 mM phosphate buffer containing 4% PFA at 4°C overnight. Slices were washed with PBS and then incubated in 10% goat serum with 0.3% Triton X-100 for 1 hr at room temperature. Sections were incubated in 4°C overnight in primary antibody for GFP (1:500 Anti-Green Fluorescent Protein Antibody Aves Labs, AB_2307313). The following day, sections were incubated in goat anti-chicken Alexa Fluor

488 secondary antibody (1:500 Abcam cat# 150169) and Alexa Fluor 594-conjugated streptavidin (1:1000 Thermo Fisher, S11227) in PBS with 0.3% Triton X-100 for 2 hr at room temperature.

Slices were mounted on a glass slide using Vectashield. Sections were imaged using a Zeiss Axioscope-5 with stereo investigator (MBF Bioscience) for analysis. Cell counts, cell-type classification, and evaluation of double labeling were conducted by an experimenter blinded to treatments. Cells with compact dendritic arbors and somata with greater length than width were classified as GCs and those with wide dendritic angle, two or more primary dendrites, and greater somatic width than height were classified as SGCs (*Gupta et al., 2020*; *Afrasiabi et al., 2022*) by a trained investigator.

## Slice physiology

Seven to nine days after tamoxifen induction following EE exposure, TRAP2-ChR2/eYFP mice were euthanized under isoflurane anesthesia for preparation of horizontal brain slices (350 µm) using a Leica VT1200S Vibratome in ice-cold sucrose artificial cerebrospinal fluid (sucrose-aCSF) containing (in mM): 85 NaCl, 75 sucrose, 24 $NaHCO_3$, 25 glucose, 4 $MgCl_2$, 2.5 KCl, 1.25 $NaH_2PO_4$, and 0.5 CaCl. Slices were bisected and incubated at 32°C for 30 min in a holding chamber containing an equal volume of sucrose-aCSF and recording aCSF and were subsequently held at room temperature for an additional 30 min before use. The recording aCSF contained (in mM): 126 NaCl, 2.5 KCl, 2 $CaCl_2$, 2 $MgCl_2$, 1.25 $NaH_2PO4$, 26 $NaHCO_3$, and 10 D-glucose. All solutions were saturated with 95% $O_2$ and 5% $CO_2$ and maintained at a pH of 7.4 for 2–6 hr (*Gupta et al., 2012*; *Yu et al., 2015*; *Afrasiabi et al., 2022*). Slices were transferred to a submerged recording chamber and perfused with oxygenated aCSF at 33°C. Whole-cell voltage-clamp and current-clamp recordings from GCs in the GC layer and presumed SGCs in the inner molecular layer or edge of the GC layer were performed under IR-DIC visualization with Nikon Eclipse FN-1 (Nikon Corporation) using ×40 water immersion objective. Recordings were obtained using axon instruments MultiClamp 700B amplifier (Molecular Devices). Data were low-pass filtered at 2 kHz, digitized using Axon DigiData 1400A (Molecular Devices), and acquired using pClamp11 at 10 kHz sampling frequency. Recordings were obtained using borosilicate glass microelectrodes (3–7 MΩ), pulled using Narishige PC-10 puller (Narishige Japan). Recordings were performed using K-gluconate-based internal solution (K-gluc) containing 126 mM K-gluconate, 4 mM KCl, 10 mM HEPES, 4 mM Mg-ATP, 0.3 mM Na-GTP, and 10 mM PO-creatinine or cesium methane sulfonate ($CsMeSO_4$) internal solution containing 140 mM cesium methane sulfonate, 10 mM HEPES, 5 mM NaCl, 0.2 mM EGTA, 2 mM Mg-ATP, and 0.2 mM Na-GTP (pH 7.25; 270–290 mOsm). Biocytin (0.2%) was included in the internal solution for post hoc cell identification (*Yu et al., 2015*; *Afrasiabi et al., 2022*; *Gupta et al., 2022*). Cells labeled with eYFP were visualized under epifluorescence and patched under IR-DIC using pipettes filled with K-gluc internal and held at –70 mV in current clamp. 10 ms, 10 Hz pulses of blue light ($\lambda$ =470 nm 0.9 mW) were used to optically evoke firing or inward currents to confirm ChR2/eYFP labeling. Responses to 1 s positive and negative current injections, beginning at –200 pA with 40 pA steps up to 20 sweeps, were examined to determine active and passive characteristics. Dual patch clamp recordings were obtained from pairs of labeled and unlabeled neurons. Unlabeled neurons were recorded using microelectrodes with $CsMeSO_4$ internal and held at 0 mV (glutamate reversal potential) to isolate IPSCs and –70 mV (close to GABA reversal potential) to record EPSCs. Labeled neurons, held in current clamp, were depolarized by 10 ms 500 pA pulses at 50 Hz to elicit APs to measure evoked responses in labeled or unlabeled cells held in voltage clamp. Labeled neuron pairs were tested for connectivity in both directions. In control experiments, wild-type mice were bilaterally injected with AAV5-CaMKIIa-hChR2(H134A)-EYFP (gift from Karl Deisseroth, Addgene plasmid # 26969) in the GC layer (AP –3.2 mm, ML ±2.6 mm, DV –2.8 mm). Four weeks after injection, mice were sacrificed for horizontal hippocampal slices (350 µm thick) that were prepared as detailed. Various ROIs were selected for stimulation of ChR2-expressing GCs using a Digital Mirror Device (DMD)-based pattern illuminator (Mightex Polygon 400), coupled to 473 nm blue LED (AURA light source), and controlled via TTL-based input from pClamp as detailed previously (*Proddutur et al., 2023*). Three progressively smaller circular ROIs with diameters of 110±10 µm, 55±5 µm, and 27±3 µm were activated in the GC layer. Light intensity was set at 2.6 mW. IPSCs were recorded from GCs outside the stimulated ROI using a cesium-based internal solution while holding the membrane potential at 0 mV.

sEPSCs were recorded in both labeled and unlabeled cells in slices from TRAP2-ChR2/eYFP and TRAP2-tdT mice induced with tamoxifen after EE and from C57BL6/N mice. Recordings were obtained

from a holding potential of –70 mV in voltage clamp for 5–10 min. In a subset of recordings, the sodium channel blocker TTX (1 µM) was used to block AP-dependent events. In control experiments, sEPSC IEI in aCSF was not different from the IEI recorded in GABA$_A$ receptor antagonist, SR95531 (10 µM, IEI in aCSF in s: 5.29±0.84, n=12 cells/3 mice; SR95531: 5.29±0,98, n=9 cells/3 mice, p=0.7 by Mann-Whitney U test) confirming that a majority of the synaptic events under these conditions are glutamatergic. Exclusion criteria were pre-established: Recordings were discontinued and not used if series resistance increased by >20% or if access resistance surpassed 25 MΩ. Furthermore, only cells with an initial RMP of –65 or lower were used. Post hoc biocytin immunostaining and morphologic analysis was used to definitively identify SGCs and GCs included in this study. If the cell could not be clearly identified using post hoc analysis, the cell and its associated recordings were excluded.

## Data analysis

Active and passive properties were analyzed using EasyElectrophysiology v2.6.3 (Easy Electrophysiology Ltd). AP, threshold, amplitude, half-width, and first spike latency were acquired from the first sweep in which the cell fired. fAHP, mAHP, and spike frequency adaptation were determined based on voltage response and firing in response to a 120 pA current injection. AP threshold was calculated using the first derivative method. Amplitude was calculated by the AP peak value minus baseline. Spike frequency accommodation was calculated using the divisor method, in which the ISI of the first two APs is divided by the ISI of the last two APs. First spike latency is the time from the start of a current pulse to the first AP.

AP and synaptic potential analysis were conducted using EasyElectrophysiology v2.6.3 (Easy Electrophysiology Ltd). AP kinetics were analyzed with a 200 kHz interpolation for rise time, decay time, and half-width. Decay was measured using a biexponential decay curve fit with a cutoff of 10–90% of the AP amplitude. Rise time is calculated between 10% and 90% of the AP amplitude. Half-width was calculated as the time between the two half-amplitude samples. Afterhyperpolarization values are calculated as baseline minus fAHP or mAHP. The value is the minimum point within a search region specified as 0–3 ms for fAHP and 10–50 ms for mAHP. Spontaneous EPSCs were detected and analyzed using EasyElectrophysiology threshold search algorithm, and events were confirmed by the experimenter. Any 'noise' that spuriously met trigger specifications was rejected. Cumulative probability plots in *Figure 5* were obtained using the same number of events from each cell.

Temporal correlation of sEPSCs: Synaptic event times used for temporal correlation analysis were extracted from sEPSCs recorded at a holding potential of –70 mV in low chloride internal solution. Temporal correlation of sEPSCs in dual recording sessions from cell pairs (L-L and L-U) was defined by a session-wise CCP of temporally binned data for select detection windows (MATLAB *xcorr*, Wiegand and Cowen). Temporal correlations of sEPSC event times in a large ±1 s detection window confirmed an expected high cross-correlation of events within the 100 ms central bin. A small ±10 ms detection window with 1 ms bin width resulted in too few co-occurring events. Consequently, CCPs were developed using multiple *detection windows* (±10 ms to ±1 s) with corresponding *bin durations* (21 bins within window). Temporal correlation across full session timelines was not calculated to avoid spuriously high correlation values from simultaneous absence of events in cell pairs (*Cutts and Eglen, 2014*). As an additional measure to avoid specious correlations, sessions with too few events, low event frequency, and short recording durations were not analyzed. Temporal correlations were tested using detection windows, and bin sizes were always divided into 21 equally sized bins in the window: a ±100 ms detection window with 10 ms bins (200 ms window, 21 bins aligned to sEPSC) and a ±50 ms detection window with 5 ms bins (100 ms window, 21 bins aligned to sEPSC). The shapes of the cross-correlograms generated from our datasets using previously established methods to evaluate monosynaptic connectivity (*Barthó et al., 2004*; *Senzai and Buzsáki, 2017*) paralleled that of the CCP plots (*Figure 6—figure supplement 2*) illustrating that the methods similarly capture co-dependencies between event time series.

Temporal correlation was determined if the CCP exceeded a 2SD threshold above the total mean correlation (0.15 for the 100 ms detection window and 0.10 for the 50 ms detection window). Within each detection window, 'peri-occurrence' was defined by the CCP maximum outside the center bin while 'co-occurrence' was defined by the correlation in the center bin. CCPs of sEPSC event times in L-L and L-U pairs were compared with randomly jittered event times from the same dataset to identify intrinsic correlations within the data. Session jitter was pseudorandomly selected from an event

timeline matrix (assigned to one cell from each paired cell recording session) bound by ±0.5 s across 100 iterations. The temporally jittered correlation data were then compared to the temporally aligned CCP and correlation data in the center bin (*Wiegand et al., 2016*). The ability of the sEPSC event co-occurrence to predict L-L versus L-U pairs was computed by plotting the ROC curve and calculating the area under the curve in both groups. Correlation values in the center bin of the CCPs were used to generate histograms (correlation bins from 0 to 0.5 with correlation bin widths of 0.001), which were reverse-integrated to evaluate the cumulative sum between categorized L-L sessions (true, n=7) and L-U sessions (false, n=8) rates. Cumulative sums were used to find the total AUROC as the classification performance measure using MATLAB (*hist counts*, *cumsum*, *flip*, and *trapz* functions), where 50% AUROC performance would classify L-L versus L-U by random chance.

Sample sizes were not predetermined and conformed with those employed in the field. Significance was set to $p < 0.05$, subject to appropriate Bonferroni correction. Statistical analysis was performed using GraphPad Prism 10 and MATLAB. Data were tested for normality and unpaired K-S test, unpaired Mann-Whitney, one-way ANOVA, two-way ANOVA, two-way repeated measures ANOVA, or Kruskal-Wallis followed by post hoc pairwise multiple comparisons using Holm-Sidak method or Dunn's method used as appropriate. Statistical data is reported as mean ± SEM or median (interquartile range) as appropriate.

## Acknowledgements

This work is supported by National Institutes of Health (NIH) NINDS R37NS069861, R01NS097750 to VS, NIH/NINDS F31NS124290 to LD.

---

## Additional information

### Funding

| Funder | Grant reference number | Author |
|---|---|---|
| National Institutes of Health | F31NS124290 | Laura Dovek |
| National Institutes of Health | R01NS097750 | Vijayalakshmi Santhakumar |
| National Institutes of Health | R37NS069861 | Edward Zagha Vijayalakshmi Santhakumar |

The funders had no role in study design, data collection and interpretation, or the decision to submit the work for publication.

### Author contributions

Laura Dovek, Conceptualization, Data curation, Formal analysis, Funding acquisition, Investigation, Visualization, Writing – original draft, Writing – review and editing; Mahboubeh Ahmadi, Data curation, Formal analysis, Investigation, Visualization, Methodology, Writing – review and editing; Krista Marrero, Software, Formal analysis, Visualization, Writing – original draft, Writing – review and editing; Edward Zagha, Formal analysis, Supervision, Validation, Writing – review and editing; Vijayalakshmi Santhakumar, Conceptualization, Supervision, Project administration, Writing – review and editing

### Author ORCIDs

Laura Dovek ⓘ https://orcid.org/0000-0002-9381-1886
Mahboubeh Ahmadi ⓘ https://orcid.org/0009-0000-0719-3632
Krista Marrero ⓘ https://orcid.org/0000-0003-3720-8756
Edward Zagha ⓘ https://orcid.org/0000-0001-5892-3746
Vijayalakshmi Santhakumar ⓘ https://orcid.org/0000-0001-6278-4187

### Ethics

This study was performed in strict accordance with the recommendations in the Guide for the Care and Use of Laboratory Animals of the National Institutes of Health. All of the animals were handled according to approved institutional animal care and use committee (IACUC) protocols (#32) of the

University of California Riverside. Mice were housed with littermates (up to 5 mice per cage) in a 12/12 h light/dark cycle. Food and water were provided ad libitum. Mice were euthanized under deep isoflurane anesthesia by decapitation, and every effort was made to minimize suffering.

Reviewer #1 (Public review): https://doi.org/10.7554/eLife.101428.3.sa1
Reviewer #2 (Public review): https://doi.org/10.7554/eLife.101428.3.sa2
Reviewer #3 (Public review): https://doi.org/10.7554/eLife.101428.3.sa3
Author response https://doi.org/10.7554/eLife.101428.3.sa4

## Additional files

### Supplementary files
MDAR checklist

### Data availability
No sequencing or western blots were generated in this study. All analyzed numerical data from figures 1, 2, 5 and 6 as well as from figure supplements are included in manuscript source data. Custom analysis source code is available in GitHub, copy archived at *VijiSanthakumarLab, 2025*. Source data are uploaded as Excel files and marked as source data.

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
