## [Editor Report · eLife Assessment]

This **useful** study describes distinctive characteristics of dentate gyrus granule cells and semilunar cells that are recruited during contextual memory processing. The study provides **solid** evidence to suggest mechanisms that may be involved in the recruitment of neurons into memory engrams in the dentate gyrus.

---

## [Referee Report · Reviewer #1 (Public review)]

Dovek and colleagues aimed at investigating the cellular and circuitry mechanisms underlying the recruitment of dentate gyrus neurons (including two morpho-physiologically-distinct subpopulations of excitatory cells called granular cells or GCs, and semilunar cells or SGCs) into memory representations, also known as engrams. To this end, the authors used TRAP2 mice to investigate the dentate gyrus "engram" neurons that were activated or not (i.e., labeled or not) in a non-fear-based context (mostly enriched environment or EE, but also Barnes Maze or BM).

A significant proportion of dentate gyrus neurons are labeled after EE exposure (35%) or after BM acquisition (15%). SGCs, distinguished from GCs using morphology-based classification, showed disproportionately context-dependent recruitment. Consistent with previous observations (Erwin et al., 2022), SGCs account for a third of behaviorally recruited "engram" neurons, although they represent less than 5% of excitatory neurons in the dentate gyrus.

Then, the authors compared the intrinsic physiological properties of GCs and SGCs that are recruited or not during EE. Consistent with previous observations (Williams et al., 2007, Afrasiabi et al., 2022), SGCs and GCs exhibited numerous differences (e.g., Rin, firing frequency) regardless of whether they were behaviorally activated or not. Differences in physiology between excitatory neuron subtypes might explain the preferential recruitment of SGCs. Interestingly, "engram" SGCs displayed lower values of adaptation in firing rate than non-recruited SGCs.

To examine how GCs and SGCs activated during EE are integrated into the local dentate gyrus microcircuits, the authors next performed a dual patch-clamp recording combined with wide-field optogenetics. Despite the presence of spontaneous EPSCs, no direct functional glutamatergic interconnection was observed between pairs of "engram" GCs and SGCs. In addition, although optogenetic stimulation of a large, random, population of neurons evokes IPSCs (indicating efficient lateral inhibition as in Stefanelli et al., 2016), the specific stimulation of behaviorally recruited GCs or SGCs rarely elicits IPSCs onto surrounding non-engram excitatory neurons.

To assess whether neurons recruited or not during EE receive differential glutamatergic drive, the authors recorded spontaneous excitatory inputs received by labeled and unlabeled GCs and SGCs. They observed that sEPSCs in labeled GCs and SGCs are more frequent and larger than in unlabeled GCs and SGCs, respectively.

Last, the authors investigated whether neurons (without discriminating GCs and SGCs) recruited in the same context were characterized by a higher propensity to receive temporally correlated inputs. To this end, they performed dual patch-clamp and analyzed the temporal correlation of spontaneous EPSCs received by pairs of neurons (either two dentate gyrus "engram" neurons, or one "engram" neuron and one "non-engram" neuron in an EE context). They observed that the temporal correlation of excitatory events received by pairs of engram neurons was greater than that of pairs of neurons that do not belong to the same ensemble, and that expected by chance.

Altogether, the data suggest that the context-dependent recruitment of dentate gyrus excitatory neurons, particularly SGCs is correlated to distinctive intrinsic properties and (correlated) excitatory afferent. Contrary to a leading hypothesis, the authors found no evidence that recruited neurons drive robust feedforward excitation of other engram neurons or feedback inhibition of non-engram neurons.

Strengths:

This article provides some information about the mechanisms that may be involved in the recruitment of neural ensembles that form non-fear-based memory engrams in the dentate gyrus. I find it interesting that the authors considered not only granular cells, the main population of excitatory neurons in the dentate gyrus, but also a sparse subpopulation of semilunar cells, a relatively understudied type of dentate excitatory neuron.

Weakness:

Most of the data presented are descriptive and based on correlation rather than causation.

---

## [Referee Report · Reviewer #2 (Public review)]

Summary:

The authors use the TRAP2 mouse line to label dentate gyrus cells active during and enriched environment paradigm and cut brain slices from these animals one week later to determine whether granule cells (GC) and semilunar granule cells (SGC) labelled during the exposure share common features. They particularly focus on the role of SGCs and potential circuit mechanisms by which they could be selectively embedded in the labelled assembly. The authors claim that SGCs are disproportionately recruited into IEG expressing assemblies due to intrinsic firing characteristics but cannot identify any contributing circuit connectivity motives in the slice preparation, although they claim that an increased correlation between spontaneous synaptic currents in the slice could signify common synaptic inputs as the source of assembly formation.

Strengths:

The authors chose a timely and relevant question, namely, how memory-bearing neuronal assemblies, or 'engrams', are established and maintained in the dentate gyrus. After the initial discovery of such memory-specific ensembles of immediate-early gene expressing engrams in 2012 (Ramirez et al.) this issue has been explored by several high-profile studies that have considerably expanded our understanding of the underlying molecular and cellular mechanisms, but still leave a lot of unanswered questions.

Weaknesses:

(1) The authors claim that recurrent excitation from SGCs onto GCs or other SGCs is irrelevant because they did not find any connections in 32 simultaneous recordings (plus 63 in the next experiment). Without a demonstration that other connections from SGCs (e.g. onto mossy cells or interneurons) are preserved in their preparation and if so at what rates, it is unclear whether this experiment is indicative of the underlying biology or the quality of the preparation. The argument that spontaneous EPSCs are observed is not very convincing as these could equally well arise from severed axons (in fact we would expect that the vast majority of inputs are not from local excitatory cells). The argument on line 418 that SGCs have compact axons isn't particularly convincing either given that the morphologies from which they were derived were also obtained in slice preparations and would be subject to the same likelihood of severing the axon. Finally, even in paired slice recordings from CA3 pyramidal cells the experimentally detected connectivity rates are only around 1% (Guzman et al., 2016). The authors would need to record from a lot more than 32 pairs (and show convincing positive controls regarding other connections) to make the claim that connectivity is too low to be relevant.

The authors now provide evidence that at least some synaptic connections are preserved by recruiting GC assemblies with channelrhodopsin, resulting in feedback inhibition which supports their argument.

(2) Another concern is that optogenetic GC stimulation rarely ever evokes feedback inhibition onto other cells which contrasts with both other in vitro (e.g. Braganza et al., 2020) and in vivo studies (Stefanelli et al., 2016) studies. Without a convincing demonstration that monosynaptic connections between SGCs/GCs and interneurons in both directions is preserved at least at the rates previously described in other slice studies (e.g. Geiger et al., 1997, Neuron, Hainmueller et al., 2014, PNAS, Savanthrapadian et al., 2014, J. Neurosci). The authors now provide evidence that at least some synaptic connections are preserved by stimulating a random subset of granule cells optogenetically, although it still remains unclear how the rate of connectivity compares to other studies or a live organism.

(3) Probably the most convincing finding in this study is the higher zero-time lag correlation of spontaneous EPSCs in labelled vs. unlabeled pairs. Unfortunately, the authors use spontaneous EPSCs to begin with, which likely represent a mixture of spontaneous release from severed axons, minis, and coordinated discharge from intact axon segments or entire neurons, make it very hard to determine the meaning and relevance of this finding. The authors now show the baseline EPSC rates and conventional Cross correlograms (CCG; see e.g. English et al., 2017, Neuron; Senzai and Buzsaki, 2017, Neuron) lending more support to this conclusion.

(4) Finally, one of the biggest caveats of the study is that the ensemble is labelled a full week before the slice experiment and thereby represents a latent state of a memory rather than encoding, consolidation, or recall processes. The authors acknowledge that in the discussion but they should also be mindful of this when discussing other (especially in vivo) studies and comparing their results to these. For instance, Pignatelli et al 2018 show drastic changes in GC engram activity and features driven by behavioral memory recall, so the results of the current study may be very different if slices were cut immediately after memory acquisition (if that was possible with a different labelling strategy), or if animals were re-exposed to the enriched environment right before sacrificing the animal. The authors discuss this limitation appropriately.

There are also a few minor issues limiting the extent of interpretations of the data:

(1) Only about 7% of the 'engram' cells are re-activated one week after exposure (line 147), it is unclear how meaningful this assembly is given the high number of cells that may either be labelled unrelated to the EE or no longer be part of the memory-related ensemble.

(2) Line 215: The wording '32 pairwise connections examined' suggests that there actually were synaptic connections; would recommend altering the wording to 'simultaneously recorded cells examined' to avoid confusion.

---

## [Referee Report · Reviewer #3 (Public review)]

Summary:

The study explores the cellular and circuit features that distinguish dentate gyrus semilunar granule cells and granule cells activated during contextual memory formation. The authors tag memory and enriched environment-activated dentate granule cells and semilunar granule cells and show their reactivation in an appropriate context a week later. They perform patch clamp recordings from activated and surrounding neurons to understand the cellular driving of the selective activation of semilunar granule cells and granule cells. Authors perform dual patch clamp recordings from various pairs of labeled semilunar granule cells, labeled granule cells, unlabeled granule cells, and unlabeled semilunar granule cells. The sustained firing of semilunar granule cells explained their preferential activation. In addition, activated neurons received correlated inputs.

Strengths:

The authors confirmed the engram cell properties of activated semilunar granule cells and granule cells in two different paradigms, validating these findings using an enriched environment paradigm.

The authors carefully separate semilunar granule cells from granule cells, using electrophysiology and morphology. Cell filling to confirm morphology further strengthens confidence.

The dual patch recordings, which are technically challenging, are carefully performed, and the presence of synaptic activity is confirmed.

The authors report that sEPSCs recorded from labelled sGCS are more frequent, higher in amplitude, and temporally correlated than their counterparts.

The authors provide evidence that lateral inhibition is not playing a role in the selective activation of sGCs during contextual learning.

Exclusive use of slice physiology limits some of these conclusions due to the shearing of connections during the slicing process.

---

## [Author Response]

The following is the authors’ response to the previous reviews

**Reviewer #1 (Public Review):**
(1) I think the article is a little too immature in its current form. I'd recommend that the authors work on their writing. For example, the objectives of the article are not completely clear to me after reading the manuscript, composed of parts where the authors seem to focus on SGCs, and others where they study "engram" neurons without differentiating the neuronal type (Figure 5). The next version of the manuscript should clearly establish the objectives and sub-aims.

We now provide clarification for focusing on the labeling status versus the cell types in figure 5. Since figure 5 focuses on inputs to labeled pairs versus Labeledunlabeled pairs the pairs include mixed groups with GCs and SGCs. Since the question pertains to inputs rather than cell types, we did not specifically distinguish the cell types. This is now explained in the text on page 15: “Note that since the intent was to determine the input correlation depending on labeling status of the cell pairs rather than based on cell type, we do not explicitly consider whether analyzed cell pairs included GCs or SGCs.”

(2) In addition, some results are not entirely novel (e.g., the disproportionate recruitment as well as the distinctive physiological properties of SGCs), and/or based on correlations that do not fully support the conclusions of the article. In addition to re-writing, I believe that the article would benefit from being enriched with further analyses or even additional experiments before being resubmitted in a more definitive form.

We now indicate the data comparing labeled versus unlabeled SGCs is novel. Moreover, we also highlight that (1) recruitment of SGCs has not been previously examined in Barnes Maze or Enriched Environment, (2) that our unbiased morphological analysis of SGC recruitment is more robust than subsampling of recorded neurons in prior studies and (3) that our data show that prior may have overestimated SGC recruitment to engrams. Thus, the data characterized as “not novel” are essential for appropriate analysis of behaviorally tagged neurons which is the thrust of our study.

**Reviewer #2 (Public Review):**
(1) The authors conclude that SGCs are disproportionately recruited into cfos assemblies during the enriched environment and Barnes maze task given that their classifier identifies about 30% of labelled cells as SGCs in both cases and that another study using a different method (Save et al., 2019) identified less than 5% of an unbiased sample of granule cells as SGCs. To make matters worse, the classifier deployed here was itself established on a biased sample of GCs patched in the molecular layer and granule cell layer, respectively, at even numbers (Gupta et al., 2020). The first thing the authors would need to show to make the claim that SGCs are disproportionately recruited into memory ensembles is that the fraction of GCs identified as SGCs with their own classifier is significantly lower than 30% using their own method on a random sample of GCs (e.g. through sparse viral labelling). As the authors correctly state in their discussion, morphological samples from patch-clamp studies are problematic for this purpose because of inherent technical issues (i.e. easier access to scattered GCs in the molecular layer).

We now clarify, on page 9, that a trained investigator classified cell types based on predefined morphological criteria. No automated classifiers were used to assign cell types in the current study.

(2) The authors claim that recurrent excitation from SGCs onto GCs or other SGCs is irrelevant because they did not find any connections in 32 simultaneous recordings (plus 63 in the next experiment). Without a demonstration that other connections from SGCs (e.g. onto mossy cells or interneurons) are preserved in their preparation and if so at what rates, it is unclear whether this experiment is indicative of the underlying biology or the quality of the preparation. The argument that spontaneous EPSCs are observed is not very convincing as these could equally well arise from severed axons (in fact we would expect that the vast majority of inputs are not from local excitatory cells). The argument on line 418 that SGCs have compact axons isn't particularly convincing either given that the morphologies from which they were derived were also obtained in slice preparations and would be subject to the same likelihood of severing the axon. Finally, even in paired slice recordings from CA3 pyramidal cells the experimentally detected connectivity rates are only around 1% (Guzman et al., 2016). The authors would need to record from a lot more than 32 pairs (and show convincing positive controls regarding other connections) to make the claim that connectivity is too low to be relevant.

We have conducted additional control experiments (detailed in response to Editorial comment #3), in which we replicated the results of Stefanelli et al (2016) identifying that optogenetic activation of a focal cohort of ChR2 expressing granule cells leads to robust feedback inhibition of adjacent granule cells. These control experiments demonstrate that the slice system supports the feedback inhibitory circuit which requires GC/SGC to hilar neuron synapses.

(3) Another troubling sign is the fact that optogenetic GC stimulation rarely ever evokes feedback inhibition onto other cells which contrasts with both other in vitro (e.g. Braganza et al., 2020) and in vivo studies (Stefanelli et al., 2016) studies. Without a convincing demonstration that monosynaptic connections between SGCs/GCs and interneurons in both directions is preserved at least at the rates previously described in other slice studies (e.g. Geiger et al., 1997, Neuron, Hainmueller et al., 2014, PNAS, Savanthrapadian et al., 2014, J. Neurosci), the notion that this setting could be closer to naturalistic memory processing than the in vivo experiments in Stefanelli et al. (e.g. lines 443-444) strikes me as odd. In any case, the discussion should clearly state that compromised connectivity in the slice preparation is likely a significant confound when comparing these results.

We have conducted additional control experiments (detailed in response to Editorial comment #3), in which we replicated the results of Stefanelli et al identifying that optogenetic activation of a focal cohort of ChR2 expressing granule cells leads to robust feedback inhibition of adjacent granule cells. These control experiments demonstrate that the slice system in our studies support the feedback inhibitory circuit detailed in prior studies. We also clarify that Stefanelli study labeled random neurons and did not examine natural behavioral engrams and discuss (on page 20) the correspondence/consistency of our results with that of Braganza et al 2020.

(4) Probably the most convincing finding in this study is the higher zero-time lag correlation of spontaneous EPSCs in labelled vs. unlabeled pairs. Unfortunately, the fact that the authors use spontaneous EPSCs to begin with, which likely represent a mixture of spontaneous release from severed axons, minis, and coordinated discharge from intact axon segments or entire neurons, makes it very hard to determine the meaning and relevance of this finding. At the bare minimum, the authors need to show if and how strongly differences in baseline spontaneous EPSC rates between different cells and slices are contributing to this phenomenon. I would encourage the authors to use low-intensity extracellular stimulation at multiple foci to determine whether labelled pairs really share higher numbers of input from common presynaptic axons or cells compared to unlabeled pairs as they claim. I would also suggest the authors use conventional Cross correlograms (CCG; see e.g. English et al., 2017, Neuron; Senzai and Buzsaki, 2017, Neuron) instead of their somewhat convoluted interval-selective correlation analysis to illustrate codependencies between the event time series. The references above also illustrate a more robust approach to determining whether peaks in the CCGs exceed chance levels.

We have included data on sEPSC frequency in the recorded cell pairs (Supplemental Fig 4) and have also conducted additional experiments and present data demonstrating that labeled cell show higher sEPSC frequency and amplitude than corresponding unlabeled cells in both cell types (new Fig 5). We also include data from new experiments to show that over 50% of the sEPSCs represent action potential driven events (Supplemental fig 3).

We thank the reviewer for the suggestion to explore alternative methods of analyses including CCGs to further strengthen our findings. We have now conducted CCGs on the same data set and report that “The dynamics of the cross-correlograms generated from our data sets using previously established methods to evaluate monosynaptic connectivity (Bartho et al., 2004; Senzai and Buzsaki, 2017) parallelled that of the CCP plots (Supplemental Fig. 6) illustrating that the methods similarly capture co-dependencies between event time series. We note, here, that while the CCG and CCP are qualitatively similar, the magnitude of the peaks were different, due to the sparseness of synaptic events.

(5) Finally, one of the biggest caveats of the study is that the ensemble is labelled a full week before the slice experiment and thereby represents a latent state of a memory rather than encoding consolidation, or recall processes. The authors acknowledge that in the discussion but they should also be mindful of this when discussing other (especially in vivo) studies and comparing their results to these. For instance, Pignatelli et al 2018 show drastic changes in GC engram activity and features driven by behavioral memory recall, so the results of the current study may be very different if slices were cut immediately after memory acquisition (if that was possible with a different labelling strategy), or if animals were re-exposed to the enriched environment right before sacrificing the animal.

As noted by the reviewer, we fully acknowledge and are cognizant of the concern that slices prepared a week after labeling may not reflect ongoing encoding. Although our data show that labeled cells are reactivated in higher proportion during recall, we have discussed this caveat and will include alternative experimental strategies in the discussion.

**Reviewer #3 (Public Review):**
(1) Engram cells are (i) activated by a learning experience, (ii) physically or chemically modified by the learning experience, and (iii) reactivated by subsequent presentation of the stimuli present at the learning experience (or some portion thereof), resulting in memory retrieval. The authors show that exposure to Barnes Maze and the enriched environment-activated semilunar granule cells and granule cells preferentially in the superior blade of the dentate gyrus, and a significant fraction were reactivated on re-exposure. However, physical or chemical modification by experience was not tested. Experience modifies engram cells, and a common modification is the Hebbian, i.e., potentiation of excitatory synapses. The authors recorded EPSCs from labeled and unlabeled GCs and SGCs. Was there a difference in the amplitude or frequency of EPSCs recorded from labeled and unlabeled cells?

We have included data on sEPSC frequency in the recorded cell pairs (Supplemental Fig 4) and have also conducted additional experiments and report and present data demonstrating that labeled cell show higher sEPSC frequency and amplitude than corresponding unlabeled cells in both cell types (new Fig 5). We also include data from new experiments to show that over 50% of the sEPSCs represent action potential driven events (Supplemental fig 3).

(2) The authors studied five sequential sections, each 250 μm apart across the septotemporal axis, which were immunostained for c-Fos and analyzed for quantification. Is this an adequate sample? Also, it would help to report the dorso-ventral gradient since more engram cells are in the dorsal hippocampus. Slices shown in the figures appear to be from the dorsal hippocampus.

We thank the reviewer for the comment. We analyzed sections along the dorsoventral gradient. As explained in the methods, there is considerable animal to animal variability in the number of labeled cells which was why we had to use matched littermate pairs in our experiments This variability could render it difficult to tease apart dorsoventral differences.

(3) The authors investigated the role of surround inhibition in establishing memory engram SGCs and GCs. Surprisingly, they found no evidence of lateral inhibition in the slice preparation. Interneurons, e.g., PV interneurons, have large axonal arbors that may be cut during slicing.Similarly, the authors point out that some excitatory connections may be lost in slices. This is a limitation of slice electrophysiology.

We have conducted additional control experiments (detailed in response to Editorial comment #3), in which we replicated the results of Stefanelli et al identifying that optogenetic activation of a focal cohort of ChR2 expressing granule cells leads to robust feedback inhibition of adjacent granule cells. These control experiments demonstrate that the slice system supports the feedback inhibitory circuit detailed in prior studies.

We now discuss (page 21) that “the possibility that slice recordings lead to underestimation of feedback dendritic inhibition cannot be ruled out.”

**Reviewer #1 (Recommendations for the authors):**
(1) I struggle to understand the added value of the Barnes Maze data (Figures 1 and S1), since the authors then focus on the EE for practical reasons. In particular, the analysis of mouse performance (presented in supplemental Figure 1) does not seem traditional to me. For example, instead of the 3 classical exploration strategies (i.e., random, serial, direct), the authors describe 6, and assign each of these strategies a score based on vague criteria (why are "long corrected" and "focused research" both assigned a score of 0.5?). Unless I'm mistaken, no other classic parameters are described (e.g., success rate, latency, number of errors). If the authors decide to keep the BM results, I recommend better justifying its existence and adding more details, including in the method section. Otherwise, perhaps they should consider withdrawing it. Even if we had to use two different behavioral contexts, wouldn't it have made sense to use, in addition to the EE, the fear conditioning test, which is widely used in the study of engrams? Under these conditions (Stefanelli et al., 2016), the number of cells recruited after fear conditioning seems sufficient to reproduce the analyses presented in Figures 2-5 and determine whether or not lateral inhibition is dependent on the type of context (Stefanelli and colleagues suggest significant strong lateral inhibition during fear conditioning, whereas the data from Dovek and colleagues suggest quite the opposite after exposure to EE).

The Barnes Maze data was included to evaluate the DG ensemble activation during a dentate dependent non-fear based behavioral task. This is now introduced and explained in the results. We have now included plots of the primary latency and number of errors in finding the escape hole to confirm the improvement over time (Supplemental Fig. 1). We specifically used the BUNS analysis to evaluate the use of spatial strategy and show that by day 6, day of tamoxifen induction, the mice are using a spatial strategy for navigation. Our approach to evaluate exploration strategy is based on criteria published in Illouz et al 2016. This is now detailed in the methods on page 25. We hope that the inclusion of the supplemental data and revisions to methods and results address the concerns regarding Barnes Maze experiments.

Regarding Stefanelli et al., 2016, please note that the study adopted random labeling of neurons using a CaMKII promotor driven reporter expression which they activated during spatial exploration of fear conditioning behaviors. As such labeled neurons in the Stefanelli study were NOT behaviorally driven, rather they were optically activated. This is now clarified in the text. The main drive for our study was to evaluate behaviorally tagged neurons which is novel, distinct from the Stefanelli study, and, we would argue, more behaviorally realistic and relevant.

Additionally, the lateral inhibition observed in Stafanelli et al was in response to activation of GCs labeled by virally mediate CAMKII-driven ChR2 expression. Using a similar labeling approach, new control data presented in Supplemental fig. 3 show that we are fully able to replicate the lateral inhibition observed by Stefanalli et al. These control experiments further suggest that the sparse and distributed GC/SGC ensembles activated during non-aversive behavioral tasks may not be sufficient to elicit robust lateral inhibition as has been observed when a random population of adjacent neurons are activated. Our findings are also consistent with observations by Barganza et al., 2020. This is now Discussed on page 21.

(2) The authors recorded sEPSCs received by recruited and non-recruited GCs and SGCs after EE exposure. However, it appears that they studied them very little, apart from a temporal correlation analysis (Figure 5). Yet it would be interesting to determine whether or not the four neuronal populations possess different synaptic properties.What is the frequency and amplitude of sEPSCs in GCs and SGCs recruited or not after EE exposure? Similarly, can the author record the sIPSCs received by dentate gyrus engram and non-engram GCs and SGCs? If so, what is their frequency and amplitude?

As suggested by the editorial comment #2, we how include data on the frequency and amplitude of the sEPSCs in GCs and SGCs used in our analysis of figure 5. Given the low numbers of unlabeled SGCs and labeled GCs in our paired recordings (Supplemental Fig. 5), we choose not to use this data set for analysis of cell-type and labeling based differences in EPSC parameters. However, we have previously reported that sIPSC frequency is higher in SGCs than in GCs. Additionally, we have identified that sEPSC frequency in SGCs is higher than in GC (Dovek et al, in preprint, DOI: 10.1101/2025.03.14.643192).

To specifically address reviewer concerns, we have conducted new recorded EPSCs in a cohort of labeled and unlabeled GCs and SGCs and present data demonstrating that labeled cell show higher sEPSC frequency and amplitude than corresponding unlabeled cells in both cell types (new Fig 5). These experiments were conducted in TRAP2-tdT labeled cells which were not stable in cesium based recordings. As such we, we deferred the IPSC analysis for later and restricted analysis to sEPSCs for this study.

(3) Previous data showed that dentate gyrus neurons that are recruited or not in a given context could exhibit distinct morphological characteristics (Pléau et al. 2021) and biochemical content (Penk expression, Erwin et al., 2020). In order to enrich the electrophysiological data presented in Figure 2, could the authors take advantage of the biocytin filling to perform a morphological and biochemical comparison of the different neuronal types (i.e., GCs and SGCs recruited or not after EE)?

Thank you for this suggestion. Unfortunately, detailed morphometry and biochemical analysis on labeled and unlabeled neurons was not conducted as part of this study as our focus was on circuit differences. In our experience, unless the sections are imaged soon after staining, the sections are suboptimal for detailed morphological reconstruction and analysis. Our ongoing studies suggest that PENK is an activity marker and not a selective marker for SGCs and we are undertaking transcriptomic analysis to identify molecular differences between GCs and SGCs. We respectfully submit that these experiments are outside the scope of this study.

(4) Figures 3 and 4 show only schematic diagrams and representative data. No quantification is shown. Instead of pie charts showing the identity of each pair (which I find unnecessary), I'll use pie charts representing the % of each pair in which an excitatory or inhibitory drive was recorded (with the corresponding n).

Please note that we did not observe evoked synaptic potentials in any except one pair precluding the possibility of quantification. However, we submit that it is important for the readers to have information on the number of pairs and the types of pre-post synaptic pairs in which the connections were tested.

(5) Figure 3: Given that GCs form very few recurrences in non-pathological conditions, it hardly surprises me that they form few or no local glutamatergic connections. In contrast, this result surprises me more for SGCs, whose axons form collaterals in the dentate gyrus granular and molecular layers (Williams et al., 2007; Save et al., 2019). To control the reliability of their conditions, could the authors check whether SGCs do indeed form connections with hilar mossy cells, as has been reported in the past? To test whether this lack of interconnectivity is specific to neurons belonging to the same engram (or not), could the authors test whether or not the stimulation of labeled GCs/SGCs (via membrane depolarization or even optogenetics) generates EPSCs in unlabeled GCs?

As suggested by the reviewer, we have examined whether widefield optical activation of all labeled neurons including GCs and SGCs lead to EPSCs in unlabeled GCs (63 cells tested). However, we did not observe eEPSCs. This data is presented on page 13, (Fig 4F) in the results and discussed on page 20. Since the wide field stimulation should activate terminals and lead to release even if the axon is severed, our data suggest the glutamatergic drive from SGC to GC may be limited.

As noted above, we have demonstrated the presence of lateral inhibition consistent with data in Stefanelli et al in our new supplementary figure 3. We have also shown that sustained SGC firing upon perforant path stimulations is associated with sustained firing in hilar interneurons (Afrasiabi et al., 2022) indicating presence of the SGC to hilar connectivity in our slice preparation. Therefore, we choose not to undertake challenging 2P guided paired recording of SGCs and mossy cells adjacent to SGC axon terminals reported in Williams et al 2007 to replicate the 9% SGC to MC synaptic connections. These 2P guided slice physiology studies are outside the technical scope of our study.

(6) Figure 4: The results are relatively in contradiction with the strong lateral inhibition reported in the past (Stefanelli et al., 2016), but the experimental conditions are different in the two studies. Stimulation of a single labeled GC or SGC may not be sufficient to activate an inhibitory neuron, and for the latter to inhibit an unlabeled GC or SGC. Is it possible to measure the sIPSCs received by unlabelled neurons during optogenetic stimulation of all labelled neurons? Could the authors verify whether under their experimental conditions GCs and SGCs do indeed form connections with interneurons, as reported before? Finally, Stefanelli and colleagues (2016) suggest that lateral inhibition is provided by dendrites- targeting somatostatin interneurons. If the authors are recording in the soma, could they underestimate more distal inhibitory inputs? If so, could they record the dendrites of unlabeled neurons?

Our new control data (Supplementary Fig. 3) using an AAV mediated CAMKII promotor driven random expression of ChR2 on GCs, similar to Stefanelli et al (2016) demonstrates our ability replicate the lateral inhibition observed by Stefanalli et al. (2016). Thus, our findings more accurately represent lateral inhibition supported by a sparse behaviorally labeled cohort than findings of Stefanelli et al based on randomly labeled neurons. This is now discussed on page 22-23. We respectfully submit that dendritic recordings are outside the scope of the current study.

We also discuss the possibility that somatic recordings may under sample dendritic inhibitory inputs on page 23 “the possibility that slice recordings lead to underestimation of feedback dendritic inhibition cannot be ruled out.”

(7) Figure 5: For ease of reading, I would substantially simplify the Results section related to Figure 5, keeping only the main general points of the analysis and the results themselves. The details of the analysis strategy, and the justification for the choices made, are better placed in the Method section (I advise against "data not shown").

We thank the reviewer for the suggestion to improve accessibility of the results and have moved text related to justification of strategy and controls to the methods. We have also removed references to data not shown.

(8) Figure 5: why do the authors no longer discriminate between GCs and SGCs?

Since figure 5 focuses on inputs to labeled pairs versus labeled-unlabeled pairs the pairs include mixed groups with GCs and SGCs. Since the question pertains to inputs rather than cell types, we did not specifically distinguish the cell types. This is now explained in the text on page 15.

(9) Figure 5: I would like to know more about the temporally connected inputs and their implication in context-dependent recruitment of dentate gyrus neurons. What could be the origin of the shared input received by the neurons recruited after EE exposure? For example, do labeled neurons receive more (temporally correlated or not) inputs from the entorhinal cortex (or any other upstream brain region) than unlabeled neurons? Is there any way (e.g., PP stimulation or any kind of manipulation) to test the causal relationship between temporally correlated input and the context-dependent recruitment of a given neuron?

We appreciate the reviewer’s comments on the need to examine the source and nature of the correlated inputs to behaviorally labeled neurons. However, the suggested experiments are nontrivial as artificial stimulation of afferent fibers is unlikely to be selective for labeled and unlabeled cells. Given the complexities in design, implementation and interpretation of these experiments we respectfully submit that these are outside the scope of the current study.

**Reviewer #2 (Recommendations for the authors):**
There are a few minor issues limiting the extent of interpretations of the data:(1) Only about 7% of the 'engram' cells are re-activated one week after exposure (line 147), it is unclear how meaningful this assembly is given the high number of cells that may either be labelled unrelated to the EE or no longer be part of the memory-related ensemble.

We now discuss (page 22-23) that the % labeling is consistent with what has been observed in the DG 1 week after fear conditioning (DeNardo et al., 2019) and discuss the caveat that all labeled cells may not represent an engram.

(2) Line 215: The wording '32 pairwise connections examined' suggests that there actually were synaptic connections, would recommend altering the wording to 'simultaneously recorded cells examined' to avoid confusion.

Revised as suggested